# SEMANTIC ROUTING IN PRE-TRAINED VISION MODELS FOR ONLINE DOMAIN-INCREMENTAL LEARNING

## ABSTRACT

Learning in the real world requires models to evolve with changing environments and cope with diverse forms of distribution shift. This is especially challenging in online domain-incremental learning, where data arrive as a non-stationary stream, each sample can be seen only once, and past observations cannot be revisited. Although pre-trained models can be used to obtain strong initial representations, standard fine-tuning in this setting leads to forgetting and poor cross-domain generalization. Inspired by how the human brain organizes experiences around semantic concepts, we propose Semantic Adapters (SAD)—lightweight modules plugged on top of any frozen pre-trained vision encoder that leverage structured semantic knowledge to guide representation updates. By routing the updates toward semantic clusters rather than domains, SAD stabilizes learning while enabling fast, one-pass adaptation. To further enrich flexibility, we introduce SADLoRA, which augments heads with low-rank parameter updates within the encoder, further enhancing adaptability while maintaining efficiency. Extensive experiments across diverse domain shifts show that both SAD versions substantially reduces forgetting and accelerates adaptation. The proposed Semantic routing with targeted updation offers a simple, fast, scalable and a viable solution for robust continual adaptation in dynamic real-world scenarios.[1]

## 1 INTRODUCTION

Real-world applications need models that can continuously adapt as environments evolve, often facing diverse forms of distribution shift. Continual learning (CL) enables models to learn new tasks sequentially without forgetting previously acquired knowledge. A fundamental obstacle in CL is catastrophic forgetting (McCloskey & Cohen, 1989; French, 1999), reflecting the trade-off between stability (preserving past knowledge) and plasticity (adapting to new input). While Class-Incremental Learning (IL) introduces entirely new classes at each task, Domain-IL maintains a fixed label space but requires robustness to shifts in input distributions. The latter arises naturally in real-world scenarios, for example, an autonomous driving system must recognize the same traffic signs despite variations in geography, weather, lighting, or sensor modality. The online Domain-IL setting is especially demanding, as data arrive only once in a stream, replay is limited or absent, and updates must be fast and memory-efficient.

Learning in continual settings involves two tightly coupled aspects: shaping the underlying *representations* and adapting the *decision boundaries*, and forgetting can occur in both. To mitigate degradation in the representation space and leverage robust features, recent works increasingly rely on pre-trained or foundation models, which provide strong generalization and transfer capabilities. Recent works have begun exploring how these models can be adapted to CL settings through parameter-efficient strategies such as prompt tuning or rehearsal-based fine-tuning (Wang et al., 2022a; Zhou et al., 2025a). These strategies can be effective when the incoming stream is close to the pretraining distribution, but under substantial domain shift they tend to overfit the latest domain and interfere with previously learned generic features, amplifying recency bias and brittle generalization (Borlino et al., 2024). Moreover, most studies with pre-trained models emphasize Class-IL; Domain-IL with such models—especially in the strict online regime—remains comparatively underexplored.

---

[1]Code will be made available upon acceptance.

Domain-incremental learning with pre-trained models presents distinct challenges: preserving generic, reusable representations while keeping the classifier unbiased. A common strategy is to freeze the encoder and update only lightweight adapters, which is computationally efficient. However, when these adapters are fine-tuned sequentially on a drifting stream, they still overwrite previously learned information and suffer from forgetting. Recent methods attach domain- or task-specific prompts or modules (Gao et al., 2024; Byun et al., 2024), which effectively capture each domain's distribution. Yet, such designs treat domains in isolation, chasing domain-specific characteristics rather than consolidating transferable knowledge. By contrast, humans organize experience around semantics: an "airplane" or "helicopter" remains an aerial vehicle whether seen as a photograph, a sketch, or a painting, or under different lighting and contexts. Cognitive studies suggest that knowledge is encoded in structured semantic networks that abstract away from surface variations (Binder et al., 2009; Ralph et al., 2017). This perspective motivates a shift from domain-centric adaptation toward meaning-centric updates, reducing interference and fostering reuse across heterogeneous shifts.

Building on this motivation, we propose **semantic routing** as a practical way to reduce forgetting and improve adaptability in continual learning. Instead of attaching domain-specific modules that quickly become brittle, our approach organizes learning around meaning: semantically related classes are grouped into coherent clusters, and inputs are routed to lightweight adapter heads that specialize within each cluster. To realize this idea, we rely on Language as it serves as a powerful, high-level abstraction prior to extract the semantic concepts in the visual inputs. We instantiate Semantic Adapters (SAD)—lightweight classifier heads trained only on samples that share a semantic concept across domains. This design (1) reduces cross-domain interference by enforcing concept-consistent decision boundaries, (2) enables domain-agnostic reuse since the number of heads does not grow with new domains, and (3) keeps an ultra-small online footprint. Our starting point is (SAD) and to further remain effective under large low-level shifts—where a frozen encoder's features can drift off-manifold—we introduce SAD-LoRA. It augments SAD with targeted low-rank adapters in late encoder blocks, coupling semantic routing with localized plasticity. This provides a controlled way to recondition features while preserving the stability of the frozen backbone.

Together, these modules provide a simple yet effective mechanism for continual adaptation that is computationally efficient. We evaluate semantic routing under one of the demanding continual learning scenarios: *online domain-incremental learning*, where data arrive in a single pass, memory and compute are constrained, and distribution shifts can be severe. Extensive empirical evaluation across multiple Domain-IL datasets and benchmarks demonstrates that the semantic approach consistently improves both final and anytime accuracy compared to multiple SOTA baselines, supporting the idea that organizing updates by meaning—rather than by domain—yields more stable online Domain-IL. We also show that such targeted semantic updating also proves more robust against natural corruptions or perturbations. Beyond empirical gains, the proposed semantic routing is a viable alternative to full fine-tuning, enabling practitioners to continually leverage pre-trained models to application-specific datasets in real-world environment.

## 2 RELATED WORKS

### 2.1 CONTINUAL LEARNING

Continual learning (CL) aims to enable the models to learn on a sequence of tasks without catastrophic forgetting. A common taxonomy distinguishes *task-incremental*, *class-incremental*, and *domain-incremental* scenarios depending on whether the output space or only the input distribution shifts across tasks (Van de Ven et al., 2022). In Domain–incremental learning (Domain-IL), the class label set is fixed across all stages while the input distribution changes. Several strategies are proposed to mitigate forgetting. Replay-based methods (Buzzega et al., 2020; Prabhu et al., 2020; Aljundi et al., 2019; Isele & Cosgun, 2018; Rolnick et al., 2019) address forgetting by storing a subset of past data for replay. ER-ACE couples rehearsal with class-balanced losses (anti-conflict) to mitigate bias toward recent classes (Caccia et al., 2021). Replay reduces drift but introduces storage/privacy costs and is less aligned with online single-pass settings. Regularization methods (Aljundi et al., 2018; Chaudhry et al., 2018) such as oEWC assigns Fisher-based importance to parameters and penalizes their drift across tasks, implemented online with a running Fisher estimate (Kirkpatrick et al., 2017). SI accumulates path-integral importance that discourages changes to parameters that have contributed large loss reductions in the past (Zenke et al., 2017). Despite their simplicity, regularization methods

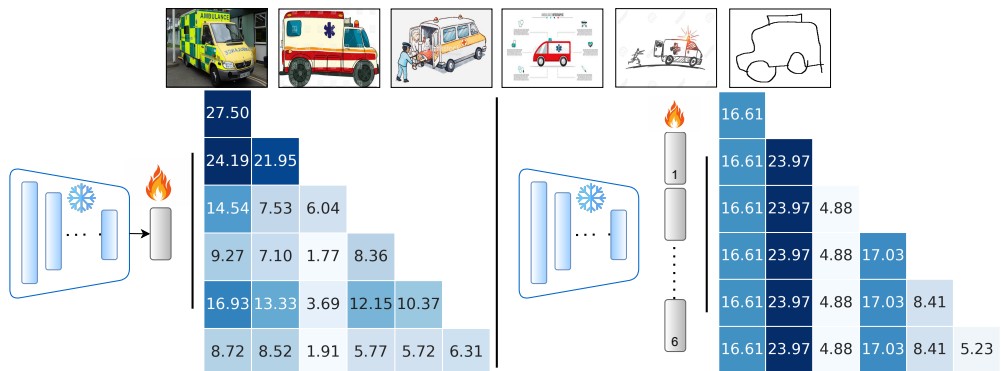

Figure 1: Online Domain-incremental learning using Pre-trained Vision Encoders on DN4IL dataset with six domains. Per-domain accuracy heatmap on - Left: a single head on all tasks (lot of forgetting). Right: domain-specific heads that require domain ID at test time (no forgetting but limited transfer). The diagonal shows in-task accuracy; off-diagonals show retention on earlier domains.

often struggle when tasks are highly dissimilar, such as in domain-incremental learning settings where input distributions shift substantially. Dynamic architectural methods (Mallya & Lazebnik, 2018; Rusu et al., 2016) tackle forgetting by expanding model capacity per task, freezing earlier weights, and learning new subnetworks.

## 2.2 PRE-TRAINED MODELS-BASED CONTINUAL LEARNING

With the advent of large pre-trained models, many fine-tuning methods minimize forgetting by leveraging the pre-trained encoder and learning small plug-ins. L2P learns a pool of prompts and retrieves a subset per input and the prompt keys act as a router in feature space (Wang et al., 2022a). DualPrompt separates general and expert prompts and trains a prompt router for better transfer (Wang et al., 2022b). There are also fine-tuning based replay techniques. SimpleCIL revisits PTM-based CIL with a streamlined recipe (light distillation, cosine heads, balanced sampling), showing strong performance with a modest buffer but still relying on stored images (Zhou et al., 2025a). MEMO shows if a small buffer can outperform a fine-tuned model without buffer and advocates memory-efficient rehearsal with careful sampling and tuning (Zhou et al., 2023). A common limitation is that many of these approaches operate in a *non-online regime*, requiring multiple epochs per domain, frequent full-batch retraining, or explicit task/domain ID information—not always feasible in truly streaming or resource-constrained settings. Moreover, prompt- or domain-specific adapters typically chase domain idiosyncrasies; when distributions keep changing, their parameters can themselves suffer from recency bias. Many works utilize pre-trained models but perform multi-stage training with full-retraining or model merging (Ainsworth et al., 2022; Matena & Raffel, 2022; Frankle et al., 2020). Our work aims to leverage the representations from a pre-trained model, and update only specific modules without updating the model.

## 3 SEMANTIC ADAPTERS

Our goal is to leverage rich representations from pre-trained vision models and perform efficient targeted adaptation in a domain-incremental setting. We begin with a simple empirical analysis on DN4IL to understand what a frozen pre-trained encoder can and cannot do under online domain shifts. We take an ImageNet-pre-trained ResNet-18, freeze all its weights, and only train classifier heads in a single pass over the six DN4IL domains (Real, Clipart, Infograph, Painting, Sketch, Quickdraw).

Figure 1 shows the accuracy on all tasks after training each task/domain. For row $t$ and column $k$, the entry "$t \rightarrow k$" is accuracy on domain $k$ after finishing stage $t$. Strong diagonals represent good performance on the current domain. The left panel shows accuracy when a single linear classifier is trained for all tasks on top of the frozen backbone. The per-domain heatmap reveals severe forgetting—new domains overwrite the decision boundary for earlier ones. The right panel replaces

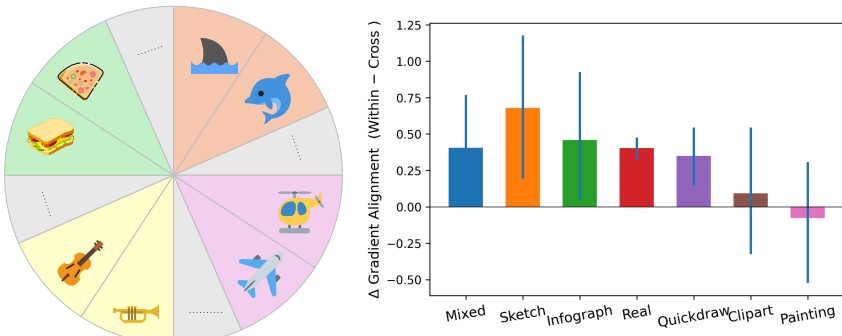

Figure 2: Semantic groups and Gradient Alignment : Within- vs cross-group gradient cosine similarity - Positive $\Delta$ indicates that updates for classes in the same semantic group align more than updates across groups, explaining why grouping by semantic meaning reduces interference.

this with domain-specific heads ; "$N$" heads for "$N$" domains (requiring domain ID at test time). This removes cross-domain interference in the classifier, largely eliminating forgetting, however, the overall accuracy remains low on several domains. Each head relearns similar concepts independently (e.g., "ambulance" in Real vs. Sketch), failing to transfer structure such as shape or relations that recur across domains.

**Motivation** Humans accumulate lifelong knowledge by organizing experiences into rich semantic abstractions rather than rote episodic traces. Classic studies of catastrophic interference show that naively updating a single network on new tasks rapidly erodes prior knowledge (McCloskey & Cohen, 1989; French, 1999). In contrast, semantic memory supports concept-level representations (e.g., "airplane," "helicopter," "hot-air balloon" as aerial vehicles; "pear," "apple," "strawberry" as fruits) that generalize across large visual changes(Tulving, 1972). Converging cognitive and neuroimaging evidence further indicates modular specialization—partially distinct subsystems tuned for faces, places, and objects—allowing stable representations with localized adaptation when new experience arrives (Kanwisher et al., 1997). Guided by this view, we aim to replace domain-conditioned heads with semantic routing.

**Semantic Groups Analysis** To test whether semantically related classes are indeed safer to share parameters, we conduct a gradient-alignment study. For the analysis we freeze a ResNet-18 encoder and train a small linear head on an anchor pair $A$ (e.g., *airplane* vs. *helicopter*). At the resulting weights $\theta_A$ we compare gradients for two new pairs: a within-group pair $B$ taken from the same semantic group as $A$ (e.g., *airplane* vs. *helicopter/UFO*), and a cross-group pair $C$ (e.g., *airplane* vs. *pizza*). We evaluate two settings: (i) per-domain, where images for $A, B, C$ are sampled from a single domain (one task) to remove style variation; and (ii) mixed, where images are pooled across all domains. For each domain/mode we summarize a compatibility score $\Delta\cos = \cos(g_A, g_B) - \cos(g_A, g_C)$, where positive values indicate that within-group updates push in more similar directions than cross-group updates. Across DN4IL (both "mixed" and "per-domain" slices), we observe consistently positive $\Delta\cos$ (Figure 2, right), meaning the gradient step for a semantically related pair is more aligned with the current task's step than the step for an unrelated pair. Thus, sharing parameters within a semantic group is cooperative, while sharing across groups creates conflicting updates. This directly motivates our routing: send each input to its semantic adapter to confine learning to a concept-consistent subspace, reducing interference and enabling targeted adaptation.

## 4 METHODOLOGY

We study Domain–Incremental Learning (Domain-IL), where the label space stays fixed while data domains change. Let $\mathcal{C} = \{c_1, \ldots, c_M\}$ be the shared class set. We train over a sequence of domains (tasks) $\{\mathcal{D}_b\}_{b=1}^{B}$, each inducing a different distribution over the same label space but with shifted input statistics. At task $b$, the training stream is $\mathcal{T}_b = \{(x_i^{(b)}, y_i^{(b)})\}_{i=1}^{N_b}$.

Figure 3: Semantic Adapters Heads (SAD) and Semantic Adapter LoRA (SADLoRA) for Pre-trained Vision encoders for Online Domain-incremental learning.

We use any frozen pre-trained vision encoder $f : \mathbb{R}^{H \times W \times 3} \to \mathbb{R}^D$ to encode the visual inputs to produce image embeddings. $z = f(x) \in \mathbb{R}^D$.

**Language–Induced Semantic Grouping:** At the beginning of the first task, we encode class labels or attributes using any light-weight pre-trained language model (PLM) and cluster them into semantic groups. This is to enable a meaningful, domain-agnostic structure on object relationships, enabling efficient routing and shared learning across heterogeneous domains.

At the beginning of Task0, we know all the classes that we will encounter in the whole training. Hence, for each class $c \in \mathcal{C}$, obtain a $d$-dimensional language embedding via a pre-trained language model (PLM),

$$e_c = \text{PLM}(c) \in \mathbb{R}^d.$$

Clustering $\{e_c\}_{c \in \mathcal{C}}$ yields $K$ disjoint semantic groups $\{\mathcal{S}_1, \dots, \mathcal{S}_K\}$ and a routing map

$$g : \mathcal{C} \to \{1, \dots, K\}, \qquad g(c) = k \iff c \in \mathcal{S}_k.$$

For each group $k$, we also save a fixed language prototype

$$L_k = \text{normalize}\left( \frac{1}{|\mathcal{S}_k|} \sum_{c \in \mathcal{S}_k} e_c \right) \in \mathbb{R}^d.$$

The semantic groups and language prototypes are computed once and remain fixed.

**Semantic Adapters on Frozen Features:** We enhance the frozen backbone with lightweight, group-specific linear classifier heads that focus on semantically coherent subsets of classes in the Domain-IL training. If there are $K$ semantic clusters formed in the previous step, we instantiate $K$ classifier heads with the corresponding classes per cluster. All the semantic adapter heads (classifiers) are instantiated once at the beginning of training. (Figure 5)

For each group $k$, the semantic adapter head is

$$A_k : \mathbb{R}^D \to \mathbb{R}^{|\mathcal{S}_k|}.$$

Given an input sample $(x, y)$, we form $z = f(x)$ from the frozen encoder and train only the *matching* head $A_{g(y)}$:

$$f_{\text{sad}} = A_{g(y)}(z) \in \mathbb{R}^{|\mathcal{S}_{g(y)}|}.$$

Let $\text{i}(y; \mathcal{S}_{g(y)})$ be the local index of $y$ inside its group. The supervised loss on the routed head is the standard cross-entropy loss,

$$\mathcal{L}_{\text{CE}} = -\log \frac{\exp\big([A_{g(y)}(z)]_{\text{i}(y;\mathcal{S}_{g(y)})}\big)}{\sum_{c \in \mathcal{S}_{g(y)}} \exp\big([A_{g(y)}(z)]_{\text{i}(c;\mathcal{S}_{g(y)})}\big)}.$$

**Language Anchoring :** Projecting visual features into the semantic language space and optimizing for alignment strengthens the semantic consistency between vision and language modalities, and proves useful for dynamic routing during inference. To encourage semantic alignment, we learn a small projection $P \in \mathbb{R}^{d \times D}$ that maps visual features to the PLM space, $\tilde{e} = \text{normalize}(P(z)) \in \mathbb{R}^d$. We apply a group-level contrastive loss:

$$\mathcal{L}_{\text{lang\_anchor}} = -\log \frac{\exp\big(\langle \tilde{e}, L_{g(y)} \rangle / \tau_\ell\big)}{\sum_{k=1}^K \exp\big(\langle \tilde{e}, L_k \rangle / \tau_\ell\big)}.$$

Inter– and Intra-Group Separation: To further discourage interference across semantic subspaces and reduce redundancy within each head, we use a separation loss that penalizes off-diagonal cosine similarity. Let $\mathcal{B}$ be a mini-batch, and for each group $k$ define the batch mean $\mu_k = \frac{1}{|\mathcal{B}_k|} \sum_{(x,y) \in \mathcal{B}_k} z$, where $\mathcal{B}_k = \{(x,y) \in \mathcal{B} : g(y) = k\}$, and $\hat{\mu}_k = \mu_k / \|\mu_k\|_2$.

$$\mathcal{L}_{\text{sep}} = \frac{1}{K(K-1)} \sum_{k \neq k'} \big\langle \hat{\mu}_k, \hat{\mu}_{k'} \big\rangle^2.$$

Similarly we also apply a similar *intra-group* de-correlation on classifier weights to make the classes inside each semantic head more distinct.

Overall Objective:

$$\mathcal{L} = \mathcal{L}_{\text{CE}} + \lambda_{\text{lang}} \mathcal{L}_{\text{lang\_anchor}} + \lambda_{\text{sep}} \mathcal{L}_{\text{sep}}.$$

**SADLoRA** In addition to training the semantic heads, we enable group-conditioned low-rank updates inside the frozen encoder to recondition features under strong shifts without full fine-tuning. SADLoRA addresses retains semantic routing and head-local learning, and add targeted low-rank plasticity in the encoder tail. This targeted plasticity—activated via semantic routing—yields robustness to large domain shifts while keeping trainables orders of magnitude below full fine-tuning. Let $\theta$ denote the backbone weights (frozen) and $\Delta\theta^{(k)}$ be group-specific low-rank updates (LoRA) inserted into a subset of layers:

$$W \mapsto W + A^{(k)} B^{(k)}, \qquad A^{(k)} \in \mathbb{R}^{m \times r}, \ B^{(k)} \in \mathbb{R}^{r \times n}, \ r \ll \min(m, n).$$

For ResNet-18 we attach LoRA to the final stage (layer4) convolutions; for ViT we attach LoRA to the last transformer block, targeting self-attention and MLP projections. We train only the group heads and the group-specific LoRA parameters in similar way as shown above.

**Inference Strategies** : Given a test sample $x$ from domain $b$, set $z = f(x)$ and $\hat{z} = z/\|z\|_2$, and let $\tilde{e} = \text{normalize}(P(z))$.

Oracle uses the true group $k = g(y)$ (upper bound), to route the sample to the correct semantic adapter and predict the output.

Dynamic Routing: In real-world we need to dynamically route without oracle knowledge. At the beginning of training, when we do semantic grouping, we also save the language prototypes per group. Further, during training we also save the group-wise visual prototypes (running means of the features within that group). From the incoming sample, we compute a routing score using visual prototypes and language prototypes by passing it through the small projection layer, we trained. We also augment this with with a head-confidence term using entropy. Language prototypes provide a domain-invariant prior when no visual statistics exist, preventing early drift. Visual prototypes capture the stream's evolving distribution and correct language mis-bias as features accumulate. Head confidence down-weights obviously wrong heads and helps recover from occasional mis-routing. Standardization across groups makes the three branches numerically commensurate and stabilizes selection early in training.

During training, for each semantic group "$g$", we maintain (i) a *language prototype* $L_g$ obtained once from a text encoder, and (ii) a *visual prototype* $V_g$ updated online as the running mean of features routed to $g$. Each group receives three scores: a visual similarity $s_g^{\text{vis}} = \langle \hat{z}, \hat{V}_g \rangle$, a language similarity $s_g^{\text{lang}} = \langle \tilde{e}, \hat{L}_g \rangle$, and a head-confidence score $s_g^{\text{head}} = -H(\text{softmax}(\phi_g(z)))$ computed from the group head $\phi_g$. We fuse all the scores $S_g$ and route to $\hat{g} = \arg\max_g S_g$ and classify with $\phi_{\hat{g}}$. More details are provided in the appendix.

Table 1: SAD results for Online Domain-IL on PTM-ResNet18 and OfficeHome and DN4IL datasets. Methods with [†] use replay.

| | Method | DN4IL | | OfficeHome | | #Trainable Params (DN4IL / Off) |
|---|---|---|---|---|---|---|
| | | $Acc$ | $AAA$ | $Acc$ | $AAA$ | |
| Scratch | SGD | $6.32_{\pm 1.86}$ | $6.23_{\pm 0.48}$ | $4.03_{\pm 0.61}$ | $3.69_{\pm 0.22}$ | 11.23M / 11.21M |
| | oEWC | $6.84_{\pm 0.32}$ | $4.76_{\pm 0.08}$ | $4.25_{\pm 0.51}$ | $3.57_{\pm 0.19}$ | |
| | SI | $7.14_{\pm 0.72}$ | $6.26_{\pm 0.32}$ | $4.61_{\pm 0.80}$ | $3.43_{\pm 0.54}$ | |
| | ER-ACE[†] | $5.12_{\pm 0.49}$ | $4.85_{\pm 0.51}$ | $3.11_{\pm 0.93}$ | $3.55_{\pm 0.52}$ | |
| PTM | SGD | $14.97_{\pm 0.58}$ | $22.32_{\pm 0.29}$ | $45.96_{\pm 0.48}$ | $37.96_{\pm 2.32}$ | 11.23M / 11.21M |
| | oEWC | $12.54_{\pm 1.35}$ | $17.95_{\pm 0.59}$ | $48.84_{\pm 2.71}$ | $41.29_{\pm 2.40}$ | |
| | SI | $17.97_{\pm 1.06}$ | $24.99_{\pm 1.52}$ | $46.72_{\pm 1.43}$ | $38.50_{\pm 1.91}$ | |
| | ER-ACE[†] | $19.69_{\pm 0.53}$ | $24.31_{\pm 0.49}$ | $50.30_{\pm 0.59}$ | $34.55_{\pm 0.82}$ | |
| PTM Frozen | SAD (oracle) | $36.66_{\pm 0.25}$ | $44.93_{\pm 0.22}$ | $74.50_{\pm 0.04}$ | $70.33_{\pm 0.91}$ | 51K / 33K |
| | **SAD** | $\mathbf{20.45_{\pm 0.30}}$ | $\mathbf{26.09_{\pm 0.30}}$ | $\mathbf{60.32_{\pm 0.30}}$ | $\mathbf{57.71_{\pm 0.42}}$ | **313K / 296K** |

## 5 EXPERIMENTAL SETUP

We begin our initial empirical analysis with the DN4IL dataset (Gowda et al., 2024), which is a curated subset of the DomainNet (Peng et al., 2019) dataset used in Domain Adaptation and considers the ease of benchmarking for continual learning purposes, and has six domains/ We also benchmark on CORe50 (Lomonaco & Maltoni, 2017), which has eleven domains and Office-Home, which has four domains (Venkateswara et al., 2017). Unless stated, results are averaged over 3 seeds. We train with only 1 EPOCH to satisfy ONLINE Domain-incremental learning evaluation. We use ImageNet-1K (Russakovsky et al., 2015) pre-trained ResNet-18 and ViT-B/16 as frozen feature extractors. The clusters and language prototypes are built with the CLIP text encoder (ViT-B/32;). We form $K$ semantic groups per dataset by clustering class text features: DN4IL($K = 16$), OfficeHome ($K = 20$), CORe50($K = 10$). For example, in DN4Il - airplanes, helicopters, flying saucers classes get clustered together as they all can be categorized as "aerial vehicles". Other clusters characterized - fruts, indoor objects, outdoor scenes, clothing etc.

We report on multiple baselines, starting with few online CL methods - oEWC (Kirkpatrick et al., 2017), SI (Zenke et al., 2017) and exemplar based ER-ACE (Caccia et al., 2021). We also show comparison with explicit PTM-based CL methods, MEMO (Zhou et al., 2023), SimpleCIL (Zhou et al., 2025a) and prompt-based methods L2P (Wang et al., 2022a), DualPrompt (Wang et al., 2022b). We also report multiple variants of our method : **SAD-Oracle**: evaluated with the oracle group-id; **SAD** : Semantic Heads evaluated with dynamic routing. We compare results with **SADLoRA** variant: SAD with *low-rank adapters* - at the end of the Results section.

**Metrics** Let $T$ be the number of tasks and $a_{t \to k}$ denote the test accuracy on task k after completing training on task t. The per-step average accuracy is $A_t = \frac{1}{T} \sum_{k=1}^{T} a_{t \to k}$ . We report $Acc$ as average performance of all tasks after the last task and Averaged Anytime Accuracy (AAA) (Caccia et al., 2021) that evaluates the model through all tasks. $Acc = A_T$ and $AAA = \frac{1}{T} \sum_{t=1}^{T} A_t$ .

## 6 RESULTS

We begin with a simple analysis on DN4IL using a pre-trained ResNet18 model(Fig. 8). We compare three setups: FineTuning with a single head, FineTuning with domain–specific heads, and SAD which freezes the encoder and trains language–derived semantic heads shared across domains (Oracle mode). SAD shows higher per–domain accuracy and a stronger overall average, whereas fine–tuning suffers severe forgetting. Task–wise heads do not suffer forgetting, but has sub-optimal performance. Grouping classes by semantics encourages the model to reuse object–level cues across domains instead of overfitting to domain style, which is crucial under single–pass learning. The significant gap in performance in SAD-Oracle proves that if routed properly, the semantic modules can adapt well to improve plasticity on new tasks while also maintaining stablility over previous tasks.

Table 2: Comparison with PTM-based methods that run longer epochs. Methods with $^\dagger$ use replay.

| Method | Office-Home | | CORe50 | | Epochs |
|---|---|---|---|---|---|
| | Acc | AAA | Acc | AAA | |
| MEMO$^\dagger$ [ICLR'23] | $63.1_{\pm1.80}$ | $71.2_{\pm2.76}$ | $68.2_{\pm2.7}$ | $66.0_{\pm2.7}$ | 20 |
| L2P [CVPR'22] | $80.0_{\pm1.29}$ | $79.7_{\pm4.19}$ | $81.7_{\pm0.4}$ | $72.3_{\pm1.1}$ | 10 |
| DualPrompt [ECCV'22] | $79.1_{\pm0.14}$ | $77.2_{\pm3.10}$ | $77.7_{\pm1.0}$ | $71.0_{\pm4.5}$ | 10 |
| SimpleCIL [IJCV'24] | $75.7_{\pm0.00}$ | $75.7_{\pm5.03}$ | $67.2_{\pm0.0}$ | $62.5_{\pm1.6}$ | 20 |
| SAD-Oracle | $88.2_{\pm0.13}$ | $85.6_{\pm0.21}$ | $90.1_{\pm0.3}$ | $86.7_{\pm0.16}$ | 1 |
| SAD | $80.1_{\pm0.08}$ | $77.8_{\pm0.8}$ | $84.8_{\pm0.56}$ | $81.1_{\pm0.46}$ | 1 |

Table 3: SAD vs. SADLoRA on Office-Home and CORe50

| | Method | ResNet18 | | | | ViT-B/16 | | | |
|---|---|---|---|---|---|---|---|---|---|
| | | Office-Home | | CORe50 | | Office-Home | | CORe50 | |
| | | Acc | AAA | Acc | AAA | Acc | AAA | Acc | AAA |
| Oracle | SAD | 74.50 | 70.33 | 72.74 | 67.81 | 88.21 | 85.63 | 90.10 | 86.70 |
| | SADLoRA | 81.20 | 73.12 | 81.66 | 78.51 | 89.24 | 86.38 | 91.85 | 87.46 |
| Dynamic | SAD | 60.32 | 57.71 | 60.35 | 53.59 | 80.14 | 77.83 | 84.82 | 81.06 |
| | SADLoRA | 69.62 | 64.21 | 72.52 | 67.45 | 81.80 | 78.16 | 85.78 | 82.09 |

Table 1 compares online Domain-IL on DN4IL and OfficeHome against few online CL baselines. We compare three training regimes: Scratch (randomly-initialized encoder), PTM (full fine-tuning of the ImageNet encoder), and PTM-Frozen (ours), which keeps the encoder fixed and trains only semantic heads. We compare against two regularization methods—oEWC and SI (which use importance-weighted regularization to slows weight drift)—and a replay method, ER-ACE (which does replay with asymmetric cross entropy to reduce bias). We report SAD (oracle)—an upper bound using the true semantic group for routing—and SAD with dynamic routing. While PTM baselines update the entire $11.2M$ parameter encoder each step, SAD trains only $300K$ parameters ($>30x$ fewer) and yet attains competitive final accuracy on both datasets. Our semantic router and adapters can utilize the rich, transferable features from the frozen encoders, and only update specialized adapters—preserving general features, limiting forgetting, and letting each head specialize to its slice of the space.

Table 2 extends the study to ViT-B/16 architecture on OfficeHome and CORe-50 datasets. Here, we compare against methods specifically proposed for PTM-based CL, although they are not online and were proposed for Class-IL settings. We report the exemplar based MEMO, adapter based SimpleCIL and prompt-based L2P and DualPrompt. These methods all train the PTM for 10–20 epochs and depend on extra resources (buffers or prompt pools) and some also retrain the whole model. SAD is competitive or better on both the datasets and in the stricter 1 epoch regime. Notably, our model updates $290K$ trainable parameters versus $82M$ parameters in the backbone. Hence the frozen PTM features + semantically routed heads yield strong generalization and proves that targeted plasticity is better than global update by fine-tuning the whole model.

**Adding low-rank plasticity.** Table 3 evaluates SAD-LoRA, which introduces low-rank updates in the encoder tail along with the semantic heads. Across Office-Home and CORe50 with both ResNet-18 and ViT-B/16, SAD-LoRA consistently improves both Oracle and Dynamic settings, with a modest increase in trainable parameters (Table 9), still much lesser than retraining any encoder. The gains are consistently larger on ResNet-18 than on ViT-B/16 as placing LoRA in the final blocks aligns especially well with CNNs, where late layers carry task-specific cues. SAD-LoRA concentrates updates where they fix domain shift without destabilizing earlier, reusable features, yielding bigger returns for the CNN architecture while still giving consistent, smaller boosts for transformers. When domain shift is large, a small amount of targeted backbone plasticity complements the semantic heads and further stabilizes performance across the stream.

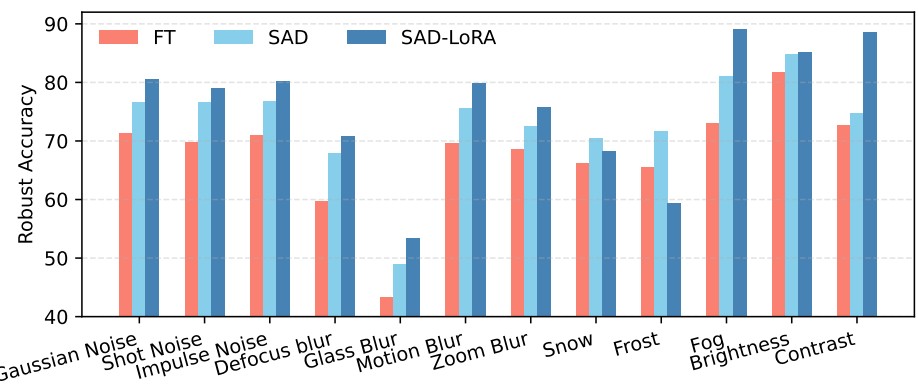

Figure 4: Robustness against different natural corruptions on ViT-B/16 and Office-home dataset

**Robustness Analysis**   To assess deployment readiness under real-world degradations, we evaluate robustness to natural corruptions on Office-Home . We apply 12 corruption types (Hendrycks & Dietterich, 2019) at severity 3 to the "real" domain test set. Across all corruptions (Figure 4), SAD consistently outperforms full fine-tuning, and SAD-LoRA yields the strongest results—especially on blur, fog, and contrast—indicating that a small, targeted capacity inside the backbone helps recondition features without destabilizing shared representations. Semantic concepts act as high-level anchors: by routing features through concept-aligned heads, the model bases decisions on stable structure (shape/parts/relations) rather than fragile pixel cues, making it less sensitive to noise or blur.

**Computational Efficiency**   Our goal was to keep the backbone encoder (CNN or Transformer) almost entirely frozen during continual learning, training only lightweight semantic heads and—under SAD-LoRA—low-rank adapters in the tail. Table 9 details the budget across datasets/architectures. For example, on the Office-Home dataset using ResNet-18, all semantic heads collectively have around 33,345 parameter, plus an additional projection layer of 262,656 parameters used for aligning vision and language embeddings to route inputs dynamically. This brings the total trainable parameters to roughly 296,000, which is minuscule compared to baseline methods that retrain the entire backbone, which have over 11 million parameters. Notably, our approach aso completes training of heads and adapters in just one epoch per domain, enabling rapid and resource-efficient online adaptation without sacrificing accuracy, making it especially suitable for quick adaptation.

## 7 CONCLUSION

We address online Domain-incremental learning setting, where non-stationary data streams demand rapid adaptation without replay and with tight compute/memory budgets. Leveraging rich representations from pre-trained encoders is appealing, but naive fine-tuning tends to overfit new domains and erode previously learned structure. Our goal was to gain targeted plasticity for the current domain while preserving stable, reusable features from the pre-trained models. We introduced semantic routing : language-induced clusters define concept groups, and we train lightweight, group-specific adapter heads on frozen encoders. We further proposed SAD-LoRA, which adds low-rank updates only in the encoder tail to provide more targeted plasticity without sacrificing stability. Across DN4IL, Office-Home, and CORe50 datasets, SAD/SAD-LoRA exceed strong PTM-based CL baselines while training only hundreds of thousands of parameters (vs. tens of millions for full fine-tuning). These results indicate that routing by meaning reduces gradient cross-talk, while targeted plasticity confines updates to the appropriate semantic subspace, preserving generic features that transfer across domains. Semantic routing also proves robust against natural corruptions. By aligning adaptation with human-meaningful semantics and keeping updates lightweight, our approach offers a practical, resource-efficient path to continual learning under real-world distribution shifts. As a broader impact, for practitioners seeking continual adaptation of pre-trained models on application-specific datasets, SAD provide a viable alternative to full fine-tuning.

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

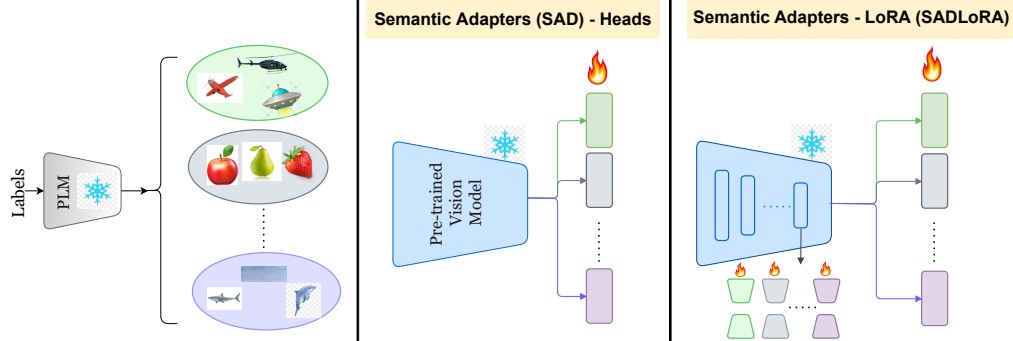

Figure 5: Semantic Adapters Heads (SAD) and Semantic Adapter LoRA (SADLoRA) for Pre-trained Vision encoders for Online Domain-incremental learning.

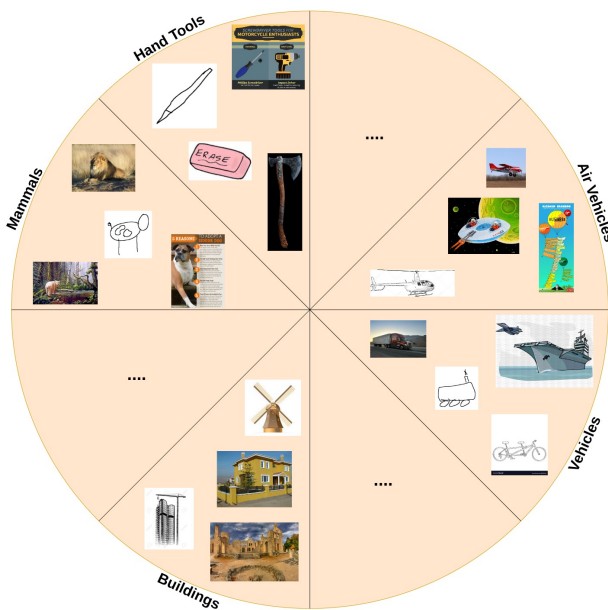

Figure 6: Conceptual clusters formed from the pretrained language model

## A   APPENDIX

## B   IDEA AND MOTIVATION

Online domain-incremental learning is a strict and realistic continual-learning regime: data arrive once, domains shift, replay is constrained or disallowed, and models must update on the fly while retaining prior competence. This setting is not addressed by many adapter-based or prompt-based methods, as their training strategy requires multiple epochs, task-specific prompt pools or domain-specialized adapters. Those approaches either depend on large prompt banks and repeated passes to stabilize, or they carve the hypothesis space by domain identity—forcing the model to relearn the same class semantics per domain and amplifying recency bias. In online Domain-IL the label set is fixed and the change is the domain; the failure mode is interference across domains for the same classes. Methods that route or adapt by task/domain keys cannot reuse class evidence accumulated elsewhere and therefore underperform when the stream drifts to unseen styles without IDs.

Our approach redefines where and how plasticity is applied. We first induce a semantic partition of the label space using language embeddings, producing a small set of concept-coherent groups. Each group

owns a lightweight classifier head; at training time, an input is routed by fused evidence—language anchors, visual prototypes accumulated online, and head confidence—so that updates land in the concept-consistent subspace most likely to transfer across domains. This yields local decision boundaries that concentrate gradient signal, reduce cross-group interference, and share parameters for the same classes over all domains. When shifts are severe, we add targeted low-rank adapters only in late backbone blocks, providing just enough feature reconditioning to handle style changes without erasing reusable representations. The result is a system that adapts in a single pass, trains hundreds of thousands (not tens of millions) of parameters, and consistently improves both final and anytime accuracy under the same budget.

## C  METHODOLOGY

**Clustering and choice of $K$.**  At Task0, we embed every class label $c \in \mathcal{C}$ with a pre-trained language model to obtain $e_c \in \mathbb{R}^d$ and $\ell_2$-normalize $z_c = e_c/\|e_c\|_2$. We form semantic groups by running $k$-means on $\{z_c\}$:

$$\{\mathcal{S}_1, \ldots, \mathcal{S}_K\} = \arg\min_{\{\mathcal{S}_k\}} \sum_{k=1}^{K} \sum_{c \in \mathcal{S}_k} \|z_c - \mu_k\|_2^2, \qquad \mu_k = \frac{1}{|\mathcal{S}_k|} \sum_{c \in \mathcal{S}_k} z_c.$$

For each group we store a fixed language prototype used later for routing.

To choose $K$, we sweep $K \in [K_{\min}, K_{\max}]$ and select the value maximizing the silhouette score:

$$\hat{K} = \arg\max_{K} \ s(K), \qquad s(K) = \frac{1}{|\mathcal{C}|} \sum_{c \in \mathcal{C}} \frac{b(c) - a(c)}{\max\{a(c), b(c)\}},$$

where $a(c)$ is the mean intra-cluster distance for class $c$ and $b(c)$ is the smallest mean distance from $c$ to any other cluster.

In practice we apply a light refinement to keep groups compact and shareable: singleton groups are merged with their nearest neighbor, oversized groups may be split with 2-means, and we enforce $K_{\min} \le K \le K_{\max}$ to avoid degenerate cases $K{=}1$ (one global head that overwrites) and $K{\approx}|\mathcal{C}|$ (no cross-class sharing). This yields concept-coherent, domain-agnostic partitions;

**Dynamic Routing (scores)**  .  Given an input $x$ from an unknown domain, extract features $z = f(x) \in \mathbb{R}^D$ and normalize $\hat{z} = z/\|z\|_2$.

Project vision to the language space using a learned map $W_p \in \mathbb{R}^{D_\ell \times D}$: $\tilde{e} = \text{normalize}(W_p z) \in \mathbb{R}^{D_\ell}$. Let $L_g \in \mathbb{R}^{D_\ell}$ be the language prototype of group $g$. Define visual and language scores:

$$s_g^{\text{vis}} = \hat{z}^\top \tilde{V}_g, \qquad s_g^{\text{lang}} = \tilde{e}^\top L_g. \tag{1}$$

Obtain head-score for each group head $\phi_g$ via entropy (lower entropy = higher confidence). Entropy measures distributional concentration (peakedness) rather than raw logit magnitude, so it's better calibrated when heads are still shallow and evolving. It reduced early mis-routes. Let $p_g = softmax(\phi_g(z))$

$$s_g^{\text{head}} = -H(p_g) = -\sum_{c \in \mathcal{C}(g)} p_g(c), \log p_g(c),, \tag{2}$$

Standardize each branch per group using running mean $\mu_g^{(\cdot)}$ and standard deviation $\sigma_g^{(\cdot)}$:

$$\bar{s}_g^{(\cdot)} = \frac{s_g^{(\cdot)} - \mu_g^{(\cdot)}}{\sigma_g^{(\cdot)} + \varepsilon}, \tag{3}$$

$$S_g = \lambda_{\text{vis}} \bar{s}_g^{\text{vis}} + \lambda_{\text{lang}} \bar{s}_g^{\text{lang}} + \lambda_{\text{head}} \bar{s}_g^{\text{head}}, \tag{4}$$

We give equal weightage to both modality prototypes 0.4 and 0.2 to the entropy score. Select the routed group as $\hat{g} = \arg\max_g S_g$.

Table 4: Online Domain-IL (1 epoch) with PTM-ViT-B/16 on OfficeHome. **"ViT-B/16" has 87M params**.

| Method | OfficeHome | # Train params |
|---|---|---|
| DUCT [CVPR'24] | 75.90 | 91M |
| DCE [ICML'25] | 64.00 | 3M |
| L2P [CVPR'22] | 77.06 | 0.1M |
| SAD-Oracle | 88.20 | |
| SADLoRA-Oracle | 89.24 | |
| SAD | 80.14 | 443K |
| SADLoRA | 81.80 | 1.9M |

# D  RESULTS

## D.1  NEW BASELINES

We also report results on more baselines in Table 4. we chose few adapter based methods that also utilize pre-trained models and report their accuracy in the online setting for fair comparison. DUCT (Zhou et al., 2025b) adapts the vision backbone by maintaining two separate branches—one to preserve prediction logits and another to preserve feature topology—which are eventually consolidated into the main model weights. However it fine-tunes and merges the massive backbone parameters directly at every task, making it significantly more computationally expensive. DCE (Li et al., 2025) adresses class and domain imbalance by training a set of frequency-aware "expert" modules alongside a dynamic selector network that routes inputs to the best expert. It requires training and storing multiple expert networks in parallel. We also report on the prompt-based L2P (Wang et al., 2022a) method which learns a pool of prompts per task.

## D.2  PLASTICITY STABILITY ANALYSIS

Continual learners must strike a balance between plasticity—learning new tasks quickly—and stability—retaining what was learned before. Following standard practice, we quantify plasticity as the average accuracy when each task is first learned (e.g., accuracy on $T2$ immediately after training on $T2$), and stability as the average accuracy on all past tasks $1 : T - 1$ after the final task $T$.

Table 5 compares four baselines—PTM with a single classifier and PTM with $N$ classifiers—each run either by fine-tuning the full encoder or by freezing the encoder and training only the heads. A key observation is that SAD uses a parameter budget comparable to task-specific heads, yet achieves substantially higher accuracy. This indicates that routing by semantic groups and adapting only the corresponding heads is far more effective for domain adaptation than naively duplicating heads per domain. Figure 7 further illustrates the plasticity–stability trade-off on DN4IL. With a frozen encoder, PTM+1 CLS shows more plasticity than stability: as a single head overwrites towards the new tasks. PTM+"N" CLS offers a small stability gain but there is less forward transfer from old tasks to next because knowledge does not transfer across domains. In contrast, SAD improves both metrics by organizing updates within concept-consistent subspaces, reducing interference and better preserving prior tasks.

## D.3  ABLATION

We provide multiple ablations. With ablation of "K", as the group count increases from a single global head toward a moderate, concept-coherent partition, performance improves. A single head (K=1) repeatedly overwrites one classifier across domains forces the model to encode all diverse semantic concepts into one dense parameter space; at the other extreme, very fine partitions leads to high granularity, resulting in adapter heads that become overly specialized to small, specific sets of concepts. This specialization severely limits positive forward transfer between semantically related but unseen classes, as the system fails to generalize learned knowledge across broader conceptual boundaries. Forward transfer (the ability to leverage knowledge acquired from previous tasks or

Table 5: Baseline performances on DN4IL ($N$=6 domains/tasks).

| Method | | DN4IL (N=6) | |
|---|---|---|---|
| | | Acc | #P |
| Trainable Encoder | PTM + 1 CLS | 14.97 | 11.2M |
| | PTM + "N" CLS | 7.79 | 11.4M |
| Frozen Encoder | PTM + 1 CLS | 6.66 | 51.3k |
| | PTM + "N" CLS | 12.69 | 300k |
| Frozen Encoder + Semantic Groups | SAD | 20.45 | 313k |
| | SAD-LoRA | 29.83 | 4.2M |

Table 6: Ablation with different "K"

| | Resnet18-Dn4IL | | | | | |
|---|---|---|---|---|---|---|
| | 1 head (all classes) | Semantic Groups | | | | |
| | K=1 | K=4 | K=8 | K=16 | K=24 | K=28 |
| Cluster Score | - | 0.0805 | 0.0863 | 0.0935 | 0.0890 | 0.0905 |
| SAD | 6.66 | 12.90 | 16.01 | **20.45** | 20.62 | 20.73 |
| SAD-LoRA | 16.17 | 20.20 | 24.05 | **29.83** | 30.03 | 30.17 |

domains to improve performance on novel, future tasks) and is an important feature for continual learning. The optimal spot is a compact set of semantically stable groups that capture shared structure without collapsing distinct concepts. Ex. in Table 13, if we reduce clusters, we need to put all animals and birds in one group perhaps and head is forced to learn highly diverse and often conflicting patterns. Or if we go very high the resulting clusters become highly specific. If the groups are too narrow, a newly encountered class (e.g., a "marker" in a new domain, which is semantically similar to "pens") won't be able to effectively reuse the knowledge learned by the "pen" adapter, effectively operating in isolation. This limits cross-domain generalization.

Across text encoders (e.g., CLIP, BERT, SBERT) and clustering strategies (e.g., k-means, Gaussian mixtures), the relative differences are small. Lightweight sentence-transformer encoders are sufficient to induce meaningful clusters, indicating that the method does not rely on a heavy language model or CLIP models. In clustering, K-means partitions data into $K$ clusters by iteratively assigning points to the nearest centroid (Euclidean distance) and updating centroids while Gaussian Mixture (GMM) models data as a weighted sum of $K$ Gaussian distributions and uses Expectation–Maximization to estimate soft clusters and both generate almos similar groupings. A second set of ablations disentangles the contribution of the training objectives. Language anchoring consistently strengthens routing and stabilizes learning when domains shift, while separation adds a modest but reliable margin between groups without encouraging domain specialization.

## E  EXPERIMENTAL DETAILS

**Benchmarks and Streams**  We study online Domain–IL on three standard vision benchmarks. DN4IL (Gowda et al., 2024) is a curated subset of DomainNet (Peng et al., 2019) containing six domains (Real, Clipart, Infograph, Painting, Sketch, Quickdraw) under a fixed class set. Office-Home (Venkateswara et al., 2017) has four domains (Art, Clipart, Product, Real-World), and CORe50 (Lomonaco & Maltoni, 2017) has eleven sessions commonly used as domains. Each dataset is processed as a single-pass stream: images arrive once per domain, we shuffle within a domain, and move to the next domain without revisiting previous data (unless a baseline explicitly requires replay). DN4IL images are $64\times64$. All other images are of $224\times224$ and normalized with ImageNet statistics. Results are averaged over three seeds.



Figure 7: Plasticity Stability Analysis of frozen Pretrained models trained on one-head, domain-wise heads and semantic heads

Table 7: Ablation with different text encoders and clustering algorithms

| R18-DN4IL - K=16 groups | | | |
|---|---|---|---|
| Encoder Ablation | | | |
| | CLIP | BERT | SBERT |
| SAD | 20.45 | 19.20 | 19.87 |
| SAD-LoRA | 29.83 | 27.50 | 28.38 |
| Clustering Ablation | | | |
| | Kmeans | Gaussian | |
| SAD | 20.45 | 19.98 | |
| SAD-LoRA | 29.83 | 28.96 | |

**Language-Induced Semantic Groups**    We construct meaning-centric groups once at the beginning of training. We encode class names with the CLIP text encoder (ViT-B/32). We also tried with class-level descriptions or attributes of each object, Example - "airplane" - a vehicle that has wings and engines and is capable of moving through the air. These light descriptions with attributes gave more nuances clusters. Text features are $\ell_2$-normalized and clustered with $k$-means to form $K$ disjoint semantic groups. We obtain $K=16$ for DN4IL, $K=20$ for Office-Home, and $K=10$ for CORe50. The language anchor for a group is the mean of its members' text embeddings, normalized. Group assignments and anchors remain fixed for the entire run.

**Architectures and Trainable Modules**    We use ImageNet-1K pre-trained **ResNet-18** and **ViT-B/16** as *frozen* encoders. On top, we attach one *Semantic Adapter* per group: a linear classifier from the

Table 8: Ablation with different loss functions

| R18-DN4IL | | | | |
|---|---|---|---|---|
| | Loss Ablation | | | |
| | CE | + Separation | + Lang_anchor | + Lang_anchor + Sep |
| SAD | 15.31 | 16.21 | 19.81 | 20.45 |
| SAD-LoRA | 20.64 | 22.64 | 27.98 | 29.83 |

Table 9: **Trainable parameter counts** for each architecture–dataset pair under our two variants. SAD trains only the semantic *classifiers* and the vision→language *projection* head. SAD-LoRA additionally trains the LoRA components inserted in the backbone tail. Counts are reported as absolute numbers

| Dataset | Arch | #P | Variant | Total trainable | Classifiers | Proj | LoRA |
|---------|------|-----|---------|-----------------|-------------|------|------|
| DN4IL | ResNet-18 | 11.23M | SAD | 313,956 | 51,300 | 262,656 | – |
| DN4IL | ResNet-18 | | SAD-LoRA | 4,008,548 | 51,300 | 262,656 | 3,694,592 |
| Office-Home | ResNet-18 | 11.21M | SAD | 296,001 | 33,345 | 262,656 | – |
| Office-Home | ResNet-18 | | SAD-LoRA | 3,654,721 | 33,345 | 262,656 | 3,358,720 |
| CORe50 | ResNet-18 | 11.2M | SAD | 288,306 | 25,650 | 262,656 | — |
| CORe50 | ResNet-18 | | SAD-LoRA | 1,967,666 | 25,650 | 262,656 | 1,679,360 |
| Office-Home | ViT-B/16 | 86.65M | SAD | 443,713 | 49,985 | 393,728 | – |
| Office-Home | ViT-B/16 | | SAD-LoRA | 1,185,593 | 49,985 | 393,728 | 741,888 |
| CORe50 | ViT-B/16 | 86.05M | SAD | 432,178 | 38,450 | 393,728 | – |
| CORe50 | ViT-B/16 | | SAD-LoRA | 805,426 | 38,450 | 393,728 | 373,248 |

Table 10: Training and Inference time of SAD methods on Resnet18-DN4IL.

| | ResNet18 - DN4IL | |
|---|---|---|
| | Train | Inference |
| SAD | 112.5 s | 0.95 ms |
| SAD-LORA | 201.8 s | 1.05 ms |

visual feature space ($D$=512 for ResNet-18, $D$=768 for ViT-B/16) to the classes in that group. In addition, we train a lightweight *vision→language* projection $W \in \mathbb{R}^{512 \times D}$ to align visual features with language space (parameter counts match: 262,656 for ResNet-18, 393,728 for ViT-B/16). In **SAD-LoRA**, we further activate low-rank adapters in the encoder tail: rank $r$=8, $\alpha$=16, dropout 0.0. For ResNet-18 we instrument `layer4` convolutions; for ViT-B/16 we instrument the last transformer block (QKV and MLP projections). All remaining encoder weights stay frozen.

**Training and Losses**    Unless noted, each stream is trained for **one epoch** (online). Batch size is 32. We optimize only the semantic heads, the vision→language projection, and (when enabled) LoRA parameters. For ResNet-18 we use SGD (lr 0.1); for ViT-B/16 we use AdamW (lr 0.001). The objective combines: (i) cross-entropy on the routed head; (ii) a language anchoring term that encourages the projected visual feature to score highest on its group anchor; and (iii) a group separation penalty that discourages cosine similarity between inter and intra groups. We set $\lambda_{\text{lang}}$=0.5 and $\lambda_{\text{sep}}$=0.1 and keep them fixed across datasets.

**Baselines and Their Resources**    We compare with two families. *Online CL baselines* include oEWC (Kirkpatrick et al., 2017) (online Fisher-based regularization) and SI (Zenke et al., 2017) (path-integral importance), both without replay, plus ER-ACE (Caccia et al., 2021) which uses rehearsal with asymmetric cross-entropy to reduce bias. *PTM-based CL baselines* include MEMO (Zhou et al., 2023) (memory-efficient rehearsal), L2P (Wang et al., 2022a) (prompt-pool retrieval for ViT), DualPrompt (Wang et al., 2022b) (general+task prompts with routing), and **SimpleCIL** (Zhou et al., 2025a) (streamlined PTM fine-tuning with rehearsal). For methods with buffers or prompt pools, we follow the settings in their papers; when they train for multiple epochs (10–20), we report their regime while our default remains single-epoch online.

**Metrics and Reporting**    We report **Acc** (final average over all tasks after the last domain) and AAA (Averaged Anytime Accuracy, i.e., the mean of per-time average accuracies across the stream). We repeat each experiment with three random seeds and report mean ± std. Implementation is in PyTorch. Parameter counts in the main paper and the appendix table break down trainables into classifiers, projection head, and LoRA components for full transparency.

Table 11: Training and Inference time of SAD methods on VIT B/16-OfficeHome

|          | VIT-B/16 - OfficeHome | |
|----------|-------|-----------|
|          | Train | Inference |
| SAD      | 81.3 s | 6.4 ms |
| SAD-LORA | 118.3 s | 7.1 ms |

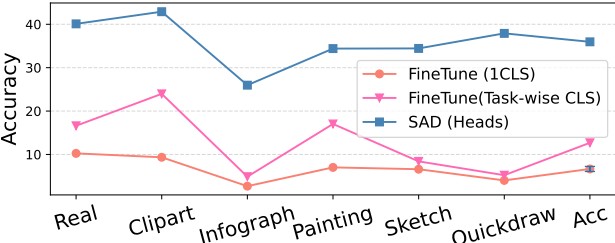

Figure 8: Accuracy on *DN4IL* dataset on Pre-trained ResNet18

## F    SEMANTIC GROUPS

We use a frozen pre-trained vision encoder (ResNet-18 or ViT-B/16) and attach lightweight *semantic adapter heads*. To define which classes share an adapter, we build *language-driven* groups: each class label is embedded with a pre-trained language model (e.g., CLIP-text or a sentence encoder), the label embeddings are Ł$_2$–normalized and clustered with $k$-means, and the resulting clusters are treated as semantically coherent groups. The group centroid serves as a fixed language prototype used for routing/analysis; only the small adapter heads are trained online. This grouping is computed once per dataset and remains fixed throughout training, ensuring a domain-agnostic, concept-level structure that encourages knowledge sharing across domains while keeping the backbone frozen.

## G    LARGE LANGUAGE MODELS USAGE IN WRITING PAPER

We used LLms to find potentially relevant prior work so we could double-check coverage of all the baselines and works related to our work. We also used it to reorganize paragraphs, improve clarity and grammar, tighten phrasing, and standardize tone (including some figure/table captions).

Table 12: Semantic groups for DN4IL.

| | |
|---|---|
| 1 | AIR VEHICLES: airplane, helicopter, flying_saucer, hot_air_balloon |
| 2 | VEHICLES: aircraft_carrier, bicycle, bus, motorbike, pickup_truck, train |
| 3 | FRUITS AND VEGGIES: apple, carrot, onion, pear, strawberry, watermelon, banana, asparagus, broccoli |
| 4 | HAND TOOLS: axe, eraser, hammer, pencil, saw, scissors, screwdriver |
| 5 | DESSERTS & FAST FOOD: birthday_cake, ice_cream, pizza, sandwich, hamburger |
| 6 | ANIMALS: bat, bird, dolphin, fish, mouse, rabbit, raccoon, squirrel, whale |
| 7 | ANIMALS (AQUATIC): dolphin, fish, shark, whale |
| 8 | MAMMALS: bear, camel, cow, dog, elephant, horse, kangaroo, lion, panda, tiger, zebra |
| 9 | INDOOR FURNITURE: bed, chair, couch, dresser, keyboard, table |
| 10 | INSECTS / ARACHNIDS: bee, butterfly, mosquito, spider |
| 11 | CLOTHING: bowtie, jacket, pants, shorts, sock |
| 12 | BUILDINGS: bridge, castle, house, skyscraper, windmill |
| 13 | NATURE / SCENES: bush, cloud, mountain, mushroom, ocean, river |
| 14 | HOUSEHOLD & MISC.: calendar, clock, cup, floor_lamp, frying_pan, map, marker, teapot, telephone, television, wine_bottle, wine_glass |
| 15 | MUSICAL INSTRUMENTS: cello, clarinet, guitar, trombone, violin |
| 16 | SEA: crab, lobster, octopus, scorpion, snail |

Table 13: Semantic groups for Office Home.

| | |
|---|---|
| 1 | WRITING TOOLS: Eraser, Pencil, Pen, Marker, Push_Pin |
| 2 | STORAGE: Shelf, File_Cabinet |
| 3 | ELECTRONICS / MEDIA: Alarm_Clock, Speaker, Radio, Webcam, Fan, Printer, TV, Monitor |
| 4 | FURNITURE: Couch, Table, Bed, Chair |
| 5 | FOOTWEAR: Sneakers, Flipflops |
| 6 | KITCHEN & TOOLS: Spoon, Knives, Paper_Clip, Scissors, Fork, Ruler, Pan, Hammer |
| 7 | OFFICE SUPPLIES: Calendar, Clipboards, Backpack, Notebook, Postit_Notes, Folder |
| 8 | POWER TOOLS: Screwdriver, Drill |
| 9 | SIGNAGE: Exit_Sign |
| 10 | COMPUTING: Computer, Keyboard, Laptop |
| 11 | TRANSPORT: Bike |
| 12 | CLEANING / HYGIENE: Mop, ToothBrush |
| 13 | CONTAINERS & MISC.: Mug, Bottle, Helmet, Soda, Trash_Can, Bucket |
| 14 | EYEWEAR: Glasses |
| 15 | OFFICE ELECTRONICS: Calculator, Telephone, Mouse |
| 16 | KITCHEN FIXTURES: Kettle, Sink |
| 17 | LIGHTING: Desk_Lamp |
| 18 | DECOR / MISC.: Flowers, Toys, Batteries |
| 19 | APPLIANCES: Refrigerator, Oven |
| 20 | HOME DECOR: Lamp_Shade, Candles, Curtains |

