# OpenReview forum: "Semantic Routing in Pretrained Vision Models for Online Domain-Incremental Learning"
_ICLR.cc/2026/Conference — ICLR 2026 Conference Desk Rejected Submission_

### Official Review · Reviewer_mE4J · 2025-10-25

**Soundness:** 2
**Presentation:** 2
**Contribution:** 2
**Rating:** 2
**Confidence:** 5

**Summary:**

This paper tackles the problem of online domain-incremental learning, where models must adapt continuously to changing environments without revisiting past data. The authors propose Semantic Adapters (SAD), a lightweight modules built on top of a frozen pre-trained vision encoder that guide updates based on semantic clusters instead of domain boundaries. They further introduce SAD-LoRA, adding low-rank adaptation to enhance flexibility. Experiments show that both methods effectively reduce forgetting, improve adaptation speed, and outperform several state-of-the-art baselines.

**Strengths:**

1. The paper is concise and presents its motivation and method in an accessible way.

2. The proposed methods achieve some strong empirical performance, surpassing some SOTA baselines.

**Weaknesses:**

1. The proposed approach largely follows a well-established paradigm in continual learning: training separate lightweight modules for different data subsets and selecting parameters during inference based on the distance between an incoming sample and prototypes. This framework has already been widely adopted in numerous prior works such as L2P, DualPrompt, and their many successors. The paper appears to only make minor procedural adjustments (e.g., limiting training to a single epoch for "online learning"), without introducing fundamentally new insights or theoretical understanding. These modifications alone are insufficient to support a strong claim of novelty.

2. The introduction of SAD-LoRA lacks clear motivation and conceptual connection to the main idea of semantic routing. It seems primarily to introduce additional trainable parameters within the encoder, which may inflate model capacity rather than reflect a meaningful methodological advancement. As a result, the performance gains might stem from increased parameters rather than the proposed semantic framework, making the comparison somewhat unfair.

3. The schematic figures are difficult to interpret and lack adequate explanatory detailsin image captions. As currently presented, they do not effectively convey the overall idea or workflow of the proposed method, which hampers reader understanding.

4. The SAD-Oracle baseline is problematic. It provides an unfair upper bound. Including it directly in the main result tables, especially near the primary baselines, can mislead readers about the practical effectiveness of the proposed method.

5. The paper does not include any ablation studies or hyperparameter sensitivity analyses, leaving unclear which components actually contribute to performance improvements. Furthermore, although the work claims to address online learning, there is no evaluation or discussion of runtime efficiency or real-time adaptability, which are essential to support this claim.

**Questions:**

See weakness.

---

> ### Author Response · Authors · 2025-11-26
>
> Thank you for your time and efforts to review our work. We recognize the improvements and have substantially revised the manuscript to address these points. A summary of updates and additional results is included in the global comment.
>
> - The contribution is not about procedural adjustments such as limiting training to a single epoch for online learning. Online Domain-IL is a distinct, stricter regime in which data are seen once, replay is unavailable, and model updates must be stable and decision-reliable without revisits. Methods like L2P/DualPrompt typically assume multi-epoch training, prompt pools, rehearsal, or task/domain IDs; these affordances let them rebalance and correct drift offline. Our contribution is a new system-level formulation for Online Domain-Incremental Learning that organizes memory by semantic concepts rather than domains. This reframes isolation in continual learning from domain-specific to concept-specific, a shift that creates concept-consistent decision regions: updates for “what it is” aggregate evidence across domains while minimizing cross-domain interference—the failure mode of unified or per-domain heads in online DIL. It is the first mechanism to use a frozen language model to enforce a concept-level structure on the vision adapter heads. The concept adapters act as specialized experts that are immediately reusable by a new domain
>
> - We introduced SAD-LoRA as a dedicated higher-capacity variant to rigorously test the boundaries of our core Semantic Adapter (SAD) methodology. The primary motivation is to validate the resilience and scalability of our concept-based routing structure when the foundational feature extractor is allowed a minimal degree of plasticity. This targeted, low-rank adaptation is a strategic choice: it acts as a mechanism to gently recondition the most abstract features when the model encounters severe domain shifts, without risking catastrophic interference to the core, generic weights. It shown in our revised tables, SAD-LoRA (1.9M trainable parameters) significantly outperforms methods like DUCT (75.90%, 91M trainable parameters) - (table9 and also in Global comment )and also fine-tuning methods (Table 5). This empirical dominance proves that the performance gains are derived not from raw capacity, but from the efficient, conceptually organized routing structure that utilizes the capacity effectively.
>
> - Thank you for the feedback. The main architecture schematic (Figure 3) has been completely redrawn and expanded, with detailed flow descriptions and extended captions. We have also corrected captions, added more explanations to the figures and analyses.
> We kept Oracle in the table to to quantify the theoretical maximum performance achievable, not to be compared to practical baselines. But we understand that it can be misleading and will arrange it better.
>
> - Ablation - We have added a comprehensive set of ablation studies - on language encoders, clustering methods, loss functions. language anchoring improves routing and stability, and results are robust across encoders and clustering.
>
> | Method   |  CLIP |  BERT | SBERT |
> | -------- | ----: | ----: | ----: |
> | SAD      | 20.45 | 19.20 | 19.87 |
> | SAD-LoRA | 29.83 | 27.50 | 28.38 |
>
> | Method   | K-means | Gaussian |
> | -------- | ------: | -------: |
> | SAD      |   20.45 |    19.98 |
> | SAD-LoRA |   29.83 |    28.96 |
>
> | Method   |    CE | + Separation | + Lang.anchor | + Lang.anchor + Sep |
> | -------- | ----: | -----------: | ------------: | ------------------: |
> | SAD      | 15.31 |        16.21 |         19.81 |               20.45 |
> | SAD-LoRA | 20.64 |        22.64 |         27.98 |               29.83 |
>
>
> - During inference, we get a sample, pass it through the encoder, obtain the feature embedding and calculate its similarity score (cosine similarity) with stored visual and language prototyes, These scores are fused to select the top semantic adapter head, which then produces the class prediction. All prototypes (visual and language) are precomputed and cached during training, so routing adds only a few dot products per group with negligible overhead. Training continually is likewise lightweight: only small adapter heads (and, in SAD-LoRA, low-rank updates in the last blocks) are updated in 1 epoch only without any replay —making both training and inference fast.

---

> > ### Author Response · Authors · 2025-11-27
> > **2/2**
> >
> > We also report the training and inference time in Table 10 and 11 and also in the global comment.
> > Both SAD variants consistently provides the fastest training and inference, confirming its suitability for continuous real-world adaptation and for low-latency deployment. (For example, DUCT method reports a running time of "2000" seconds in their paper for VIT-B/16 - Officehome)
> >
> > Example - VIT B/16 - Office home
> > | Method   | Train   | Inference |
> > | -------- | ------- | --------- |
> > | SAD      | 81.3 s  | 6.4 ms    |
> > | SAD-LORA | 118.3 s | 7.1 ms    |
> >
> > More timing details per task/domain :
> >
> > | Arch         | Dataset     | Domain    | Batch Size | Images | Total Time (ms) | Time / Image (ms) |
> > | ------------ | ----------- | --------- | ---------- | ------ | --------------- | ----------------- |
> > | vit_b_16_sad | office_home | Art       | 64         | 729    | 4081.04         | 5.5981            |
> > | vit_b_16_sad | office_home | Clipart   | 64         | 1310   | 6244.22         | 4.7665            |
> > | vit_b_16_sad | office_home | Product   | 64         | 1332   | 6374.02         | 4.7853            |
> > | vit_b_16_sad | office_home | RealWorld | 64         | 1308   | 13250.23        | 10.1306            |
> >
> > | Arch              | Dataset     | Domain    | Batch Size | Images | Total Time (ms) | Time / Image (ms) |
> > | ----------------- | ----------- | --------- | ---------- | ------ | --------------- | ----------------- |
> > | vit_b_16_sad_lora | office_home | Art       | 64         | 729    | 4594.55         | 6.3308            |
> > | vit_b_16_sad_lora | office_home | Clipart   | 64         | 1310   | 6443.76         | 4.9189            |
> > | vit_b_16_sad_lora | office_home | Product   | 64         | 1332   | 6554.50         | 4.9207            |
> > | vit_b_16_sad_lora | office_home | RealWorld | 64         | 1308   | 14747.46        | 11.2748           |

---

### Official Review · Reviewer_1HZn · 2025-10-26

**Soundness:** 2
**Presentation:** 2
**Contribution:** 3
**Rating:** 4
**Confidence:** 2

**Summary:**

This paper addresses the challenge of online domain-incremental learning (single-pass data access, no replay, and the need to adapt to distribution shifts) by proposing two lightweight semantics-based methods. The first is Semantic Adapters (SAD), which freezes pre-trained vision encoders, forms semantic clusters by clustering classes via language models, and assigns dedicated classifier heads to each cluster to enable cross-domain knowledge reuse. The second is SAD-LoRA, which inserts cluster-specific low-rank adapters into the final layers of the encoder to enhance adaptability. On datasets including DN4IL, Office-Home, and CORe50, both methods outperform baselines such as oEWC and ER-ACE.

**Strengths:**

1. Only lightweight modules are trained (with 1/30 the parameters of full fine-tuning), supporting both CNN and Transformer architectures.

**Weaknesses:**

1. Insufficient Analysis of Semantic Cluster Design: It fails to verify the impact of cluster number (K) on performance or compare the effectiveness of different clustering methods/embedding sources, leaving the optimality of cluster design unsupported.

2. Lack of Ablation on Loss Functions: It does not decompose the individual contributions of cross-entropy, language anchoring, and separation losses, nor does it justify the use of fixed loss weights (λ_lang=0.5, λ_sep=0.1).

**Questions:**

see Weaknesses.

---

> ### Author Response · Authors · 2025-11-27
>
> We thank the reviewer for their feedback and their positive assessment of our parameter efficiency and the constructive feedback regarding the ablation studies.
>
> - We now report multiple ablations in the Appendix of the paper (also please check the global comment for summary of all changes).
> - Impact of "K" - We now provide an explicit sweep over the number of semantic groups $K$ on DN4IL with ResNet-18 (Table 6). When $K=1$, the model degenerates to a single head over all classes and suffers from severe cross-domain interference. Increasing $K$ to number of classes prevents interference but minimizes Forward Transfer, as related classes (e.g., 'Couch' and 'Chair' in the 'furniture' group) can no longer share learned features.
> We explain the clustering mechanism more in paper as well. At $t=0$, we embed class information with a light weight PLM and cluster these embeddings. When we start training, we don't know the optimal number of clusters (as we dont calculate accuracy and re-iterate as it in online continual learning). So we start from $Kmin$ and $Kmax$ from 4 to 25(1/4th of classes), and report the silhouette coefficient [1], and we use this score to select the optimal K, that is optimal between relevant groups vs time for training an routing.
>
> |                   |  K=1  |   K=4  |   K=8  |  K=16  |  K=24  |  K=28  |
> | ----------------- | :---: | :----: | :----: | :----: | :----: | :----: |
> | **Cluster Score** |   —   | 0.0805 | 0.0863 | 0.0935 | 0.0890 | 0.0905 |
> | **SAD**           |  6.66 |  12.90 |  16.01 |  20.45 |  20.62 |  20.73 |
> | **SAD-LoRA**      | 16.17 |  20.20 |  24.05 |  29.83 |  30.03 |  30.17 |
>
> - To test whether performance hinges on a particular language model or clustering algorithm, we add encoder and clustering ablations (Table 7). Using CLIP, BERT, and a lightweight SBERT encoder leads to very similar trends showing that semantic grouping does not rely on a specific or heavy PLM and can be instantiated with compact sentence transformers as well. Likewise, replacing k-means with a Gaussian Mixture Model yields comparable results
>
> **R18-DN4IL – K=16 groups**
>
> |                |        | CLIP  | BERT  | SBERT |
> |----------------|--------|------:|------:|------:|
> | **Encoder Ablation** | SAD      | 20.45 | 19.20 | 19.87 |
> |                | SAD-LoRA | 29.83 | 27.50 | 28.38 |
>
> |                |          | Kmeans | Gaussian |
> |----------------|----------|-------:|--------:|
> | **Clustering Ablation** | SAD      | 20.45  | 19.98   |
> |                | SAD-LoRA | 29.83  | 28.96   |
>
>
> - We explicitly disentangle the contributions of the three loss terms on DN4IL (Table 8). Starting from cross-entropy alone, adding only the separation loss (“CE + Separation”) yields a consistent but moderate gain. Adding only the language anchoring term (“CE + Lang. anchor”) produces a larger improvement, confirming that aligning projected visual features with their group language prototypes is the main driver of better routing and invariance.
> For the hyper-parameters, we have only and we did a grid search from 0 to 2 in degrees of 0.2 to choose the best combination.
>
> | Method   |    CE | + Separation | + Lang.anchor | + Lang.anchor + Sep |
> | -------- | ----: | -----------: | ------------: | ------------------: |
> | SAD      | 15.31 |        16.21 |         19.81 |               20.45 |
> | SAD-LoRA | 20.64 |        22.64 |         27.98 |               29.83 |
>
> - To strengthen the online-learning claim, we additionally report  new comparisons with the latest adapter based methods. Table 4 in paper and also in global comment.
>
> - compute - We also include comprehensive trainable-parameter counts and per-architecture budgets (Table 9), as well as measured training and inference times (Tables 10–11). The SAD versions (both of them) are very fast - 0.9ms per image, which is extremely suitable for real-time adaptation.
>
> We hope these extended ablations and efficiency analyses address the reviewer’s concerns

---

### Official Review · Reviewer_3Yy5 · 2025-10-30

**Soundness:** 3
**Presentation:** 3
**Contribution:** 3
**Rating:** 2
**Confidence:** 4

**Summary:**

The proposed method, Semantic Adapters (SAD), addresses the challenges of catastrophic forgetting and poor cross-domain generalization in online domain-incremental learning. SAD replaces traditional domain-specific adaptation with meaning-centric updates. It operates by first using a pre-trained language model (PLM) to encode and cluster class labels into a small set of coherent semantic groups, each representing a concept. For each of these semantic groups, a lightweight classifier head (SAD head) is instantiated. During training, the visual encoder remains frozen, and inputs are dynamically routed to the appropriate SAD head based on their semantic meaning, allowing learning to concentrate within concept-consistent subspaces. This design stabilizes learning, reduces interference between domains, and promotes knowledge reuse, enabling fast, one-pass adaptation.

To further enhance adaptability, the paper introduces SAD-LORA, an extension of SAD. SAD-LORA augments the core SAD framework by incorporating low-rank adapters (LoRA) into the final blocks of the frozen pre-trained vision encoder. This combination of semantic routing with targeted low-rank plasticity allows the model to recondition features more effectively to handle significant domain shifts and corruptions, such as blur or fog, while preserving stable, reusable representations. Both SAD and SAD-LORA achieve competitive or superior performance compared to existing methods.

**Strengths:**

Semantic routing effectively mitigates catastrophic forgetting by organizing learning around stable semantic concepts rather than isolated domains, preventing interference between tasks.

The method is computationally efficient, training only lightweight semantic adapter heads and optional low-rank adapters, leading to significantly fewer trainable parameters and faster online adaptation.

**Weaknesses:**

- Lack of Core Technical Novelty: The proposed SAD method primarily combines well-established techniques such as pre-trained vision encoders, language model embeddings for semantic grouping, and lightweight adapter heads. It doesn't introduce fundamentally new algorithms or architectural innovations.


- SADLORA appears to be a re-branding of the standard LoRA (Low-Rank Adaptation) technique applied within their framework, rather than proposing a novel variant or significant extension of LoRA itself.

- The paper compares its online single-pass method (SAD) against several baselines that are allowed multiple training epochs (20) per domain or rely on replay buffers (e.g., MEMO, L2P, DualPrompt, SimpleCIL), which could provide them an unfair advantage in stability and performance.

- The method heavily leverages a CLIP text encoder for semantic grouping and language anchoring. However, it lacks direct comparisons against other recent domain-incremental learning methods that also specifically utilize CLIP's powerful cross-modal representations.

-  The paper states different LoRA placement strategies for ResNet-18 (layer4 convolutions) and ViT-B/16 (last transformer block's self-attention and MLP projections) without thoroughly justifying why these specific locations were chosen or presenting ablation studies to demonstrate their optimality.

**Questions:**

see weaknesses.

---

> ### Author Response · Authors · 2025-11-26
>
> Thanks for the thoughtful review. We’ve carefully addressed your comments and substantially revised the manuscript (revisions highlighted in blue); a summary of updates and additional results is included in the global comment.
>
> - Our contribution is in the in the novel, systemic-level design to solve the challenging problem of Online Domain-Incremental Learning (Online DIL) via a concept-driven memory organization. The design includes using a cross-modal (language) teacher to perform  unsupervised clustering of the label space, defining a fixed set of concepts that acts as a cognitive map. This process fundamentally redefines knowledge isolation in CL from domain-specific to concept-specific, a paradigm shift for DIL.
> This yields concept-consistent decision regions that aggregate evidence across domains and reduce cross-domain interference—precisely the failure mode of unified or domain-specific heads in online Domain-IL.
>
> - SAD-LoRA is presented as a higher-capacity variant of SAD. Its purpose is to demonstrate that the semantic routing structure is robust and effective even when allowing small updates to the deep backbone. It is targeted low-rank plasticity in the encoder tail, activated only to gently recondition features when there are large domains shifts. On our benchmarks, SAD-LoRA outperforms full fine-tuning while using far fewer parameters.
>
> | Setting                          | Method        | Acc   | #P    |
> | -------------------------------- | ------------- | ----- | ----- |
> | Trainable Encoder                | PTM + 1 CLS   | 14.97 | 11.2M |
> |                                  | PTM + “N” CLS | 7.79  | 11.4M |
> | Frozen Encoder                   | PTM + 1 CLS   | 6.66  | 51.3k |
> |                                  | PTM + “N” CLS | 12.69 | 300k  |
> | Frozen Encoder + Semantic Groups | SAD           | 20.45 | 313k  |
> |                                  | SAD-LoRA      | 29.83 | 4.2M  |
>
> SAD-LoRA also significantly outperforms high-capacity baselines like DUCT, while remaining $\sim 45 \times$ more parameter-efficient (1.9M vs. 91M).Table 4 in paper, also shown in global comment.
>
> - Most continual-learning methods do not focus on the strict online setting, as it is challenging to learn with a single pass over streaming data. Direct comparisons are therefore limited; Outperforming these methods thar are allowed multiple training epochs or rely on rehearsal buffers (which give them access to past data) is a major strength of SAD. We also present new results and compare against few adapter based methods in the online setting. Table 4 in paper, also shown in global comment.
>
> - We specifically did ablation on the language encoder and clustering: CLIP, BERT, and SBERT produce similar trends. This shows the grouping mechanism is not tied to a particular text backbone and even a light-weight sentence transformer ("all-MiniLM-L6-v2") can be used to encode text and used for clustering.
>
> | Method   |  CLIP |  BERT | SBERT |
> | -------- | ----: | ----: | ----: |
> | SAD      | 20.45 | 19.20 | 19.87 |
> | SAD-LoRA | 29.83 | 27.50 | 28.38 |
>
> - The specific placement of LoRA and Adapters was inspired by the the well-established hierarchy of visual features in deep neural networks [1][2][3]. The convolutional layers (ResNet early layers) or early transformer blocks (ViT) learn generic, low-level features (edges, textures) that are reusable across all domains. The later layers (ResNet layer4 or ViT's final blocks) specialize in high-level, task-specific features (object parts, holistic concepts). To balance reuse of pre-trained knowledge with modest task-specific adaptation, we therefore localize LoRA to the late blocks, focusing adaptation where distributional drift concentrates and retaining the robustness of early features.
>
> [1] Yosinski, Jason, et al. "How transferable are features in deep neural networks?."
> [2] Kornblith, Simon, Jonathon Shlens, and Quoc V. Le. "Do better imagenet models transfer better?."
> [3] He, Junxian, et al. "Towards a unified view of parameter-efficient transfer learning."

---

> ### Comment · Reviewer_3Yy5 · 2025-11-26
>
> I thank the authors for their response. It clarified some of my concerns, particularly regarding the selection of LoRA positions. I have decided to raise my score. However, I still maintain that the paper's novelty and writing quality are weak.

---

### Official Review · Reviewer_U5ug · 2025-11-01

**Soundness:** 2
**Presentation:** 1
**Contribution:** 2
**Rating:** 2
**Confidence:** 3

**Summary:**

The paper proposes an approach for online domain incremental learning by leveraging label space information with a light-weight language model. The class labels are first clustered into semantic similar groups, and the inputs are routed to their corresponding classification systems. The proposed approach also maps the image representation to the language space, and uses LoRA in encoder blocks to increase model capacity.  Empirical results on DN4IL, OfficeHome and CORe50 indicate the proposed approach improves both final and anytime accuracy compared with existing approaches (MEMO, L2P, DualPrompt, SimpleCIL).

**Strengths:**

1. I liked the idea of using language information to aid domain incremental learning, where the language representation acts as a regularizer to prevent domain overfitting.
2. The proposed approach outperforms existing approaches on DN4IL, OfficeHome and CORe50 benchmarks.

**Weaknesses:**

1. I think the paper is difficult to understand and lacks a coherent and convincing story. Many important details, such as how Dynamic Routing works exactly at line 301 are under-explained. The heatmap in figure 1 is hard to interpret and the analysis related to Fig. 2 feels superficial.

2. The impact of semantic grouping is unclear.  The paper does not provide convincing evidence or ablations demonstrating the need for semantic grouping and the potential impact of incorrect routing. It would be helpful to understand results with a single cluster and with the number of clusters set equal to the number of class labels.

**Questions:**

1. How dynamic routing works exactly?
2. What is the impact of semantic grouping? What if  there is only a single cluster or when the cluster number is set to the number of class labels.
3. What happens if an example is mis-routed and how are such cases corrected during training?
4. What does the heatmap in Fig 1 represent?

---

> ### Author Response · Authors · 2025-11-27
>
> We thank the reviewer for the feedback and their questions regarding the narrative clarity of our paper. We made significant changes in the paper to improve the content, as well as new results and analyses. All changes are in blue
>
> - Dynamic routing is the inference-time mechanism that assigns an incoming input $x$ to the most relevant semantic adapter group $\hat{g}$.  It functions by fusing three distinct scores to ensure robustness against domain shifts. (1) Visual similarity - We calculate the cosine similarity between the input's normalized visual feature and the group-wise Visual Prototypes (2) Language similarity - We project the visual feature into the language space using a learned projection $P$. We then calculate similarity against fixed Language Prototypes (3) Head confidence - We compute the negative entropy of the prediction from each group's adapter head. This fusion couples a semantics-anchored prior (language) with stream-adaptive evidence (visual prototype + entropy). More details are now provided in both main paper and in the Appendix.
>
> - This is also inspired to address misrouting ; If an input is mis-routed to a group where it does not semantically belong, that adapter head will likely output a high-entropy (uncertain) prediction. The Head Confidence score penalizes this high entropy, discouraging the router from selecting that group during training. Further, during training, language anchoring aligns features toward the correct group’s language prototype, shrinking future routing errors; simultaneously, running visual prototypes update online, improving calibration as the stream evolves—all without replay or domain IDs.
>
> - The semantic grouping ($K$) controls the trade-off between interference and knowledge transfer. We ablate this in Table 6.
> K = 1; With only one cluster, gradients from visually disparate concepts (e.g., "Airplanes" and "Broccoli") collide in the shared parameters, causing massive catastrophic forgetting.  If we increase $K$ significantly (approaching the number of classes), we approach a scenario of isolated learning. While this avoids interference, it suppresses forward transfer (e.g., airplane and helicopter no longer reinforce each other as “aerial vehicles”). In practice, a moderate, concept-coherent partition yields the best balance: groups are broad enough to support reuse across related classes, yet specific enough to limit destructive interference.
>
> Clustering - We explain the clustering mechanism more in paper as well. At $t=0$, we embed class information with a light weight PLM and cluster these embeddings. When we start training, we don't know the optimal number of clusters (as we dont calculate accuracy and re-iterate as it in online continual learning). So we start from $K_min$ and $K_max$ from 4 to 25(1/4th of classes), and report the silhouette coefficient [1]. For each class attribute, $a_c$ be the mean intra-cluster distance and $b_c$ the smallest mean distance to another cluster; the per-class silhouette is
> \begin{equation}
> s(c) = \frac{b_c - a_c}{\max{a_c,,b_c}} ;\in [-1,1],
> \end{equation}
> Higher values indicate tighter, better-separated groups. We then select a $K$  that attains strong silhouette scores while respecting compute/latency constraints: larger groups increases the number of prototypes to store and the routing fan-out at inference. This yields a middle-ground that is both accurate and budget-aware. We also provide timing information below and in Table 10 and 11
>
> |                   |  K=1  |   K=4  |   K=8  |  K=16  |  K=24  |  K=28  |
> | ----------------- | :---: | :----: | :----: | :----: | :----: | :----: |
> | **Cluster Score** |   —   | 0.0805 | 0.0863 | 0.0935 | 0.0890 | 0.0905 |
> | **SAD**           |  6.66 |  12.90 |  16.01 |  20.45 |  20.62 |  20.73 |
> | **SAD-LoRA**      | 16.17 |  20.20 |  24.05 |  29.83 |  30.03 |  30.17 |
>
> --------------------------------------------------------
> | ResNet18 – DN4IL (K=16) |   Train | Inference |
> | ---------------- | ------: | --------: |
> | SAD              | 112.5 s |   0.95 ms |
> | SAD-LoRA         | 201.8 s |   1.05 ms |
>
>
>
> [1] Peter J. Rousseeuw (1987). “Silhouettes: a Graphical Aid to the Interpretation and Validation of Cluster Analysis”. Computational and Applied Mathematics 20: 53-65.

---

> > ### Author Response · Authors · 2025-11-27
> > **2/2**
> >
> > - Figure 1 (heatmap) visualizes forgetting and transfer across domains
> >  Rows: training stages (the domain just learned);
> >  Columns: evaluation domains seen so far;
> > Rows are the training stage t (domain just learned); columns are test domains k; entry (t,k) in matrix  is accuracy on domain k after finishing stage t. We see strong diagonals (good performance on the current domain), Ex. cell (2,2) represents accuracy of 2nd task after training on 2nd task and (2,1) represents accuracy on task1 (old task) after training on task2
> > The off-diagonal terms thus show forgetting - previous domains drop as we adapt to new ones.  We show two panels: a single global head, domain-ID oracle heads. The global head exhibits severe forgetting; domain-ID heads avoid forgetting by fully isolating domains but cannot share knowledge across them. Semantic heads, in contrast, (Fig 8 final row accuracy) retain performance on earlier domains while enabling transfer between semantically related classes, striking a better balance between stability and reuse.
> >
> > - Figure 2 is the empirical test that further proves our routing design. We freeze a ResNet-18, compute the gradient of a small “anchor” classifier on pair A, and compare its cosine alignment with gradients from semantically related pairs (B, same group) versus unrelated pairs (C, different group). Δcos is consistently positive, showing that updates for semantically related classes are cooperative while those for unrelated classes are conflicting within and oustide varying domains. That directly justifies concentrating learning within concept-consistent subspaces via semantic heads in the challenging domain-adaptation scenario. It also explains why “domain heads” help forgetting yet underperform overall (they avoid cross-domain interference but fail to reuse structure like shape/parts across domains), which we also show in Fig. 1
> >
> > - We have also added another analysis to measure Plasticity stability trade-off in Figure 7. We quantify plasticity as the average accuracy when each task is first learned, and stability as the average accuracy on all past tasks after training the final task. The comparison shows that PTM baselines either favor plasticity at the expense of severe forgetting or over-stabilize and stop improving on new domains, whereas SAD and especially SAD-LoRA improve both axes simultaneously.
> >
> > - Story - Our core motivation is to tackle a fundamental challenge in Online Domain-Incremental Learning: how to maintain long-term memory and enable forward transfer when visual appearance keeps shifting (domain shift) and past data cannot be stored or revisited. Our key idea is to organize memory around stable, high-level semantic concepts rather than domain IDs. The Semantic Router is, to our knowledge, the first mechanism that uses language-derived anchors to guide feature processing in this setting. This yields stability by isolating unrelated concepts (reducing catastrophic interference) and promotes forward transfer by allowing semantically related classes to share adapter capacity, leading to strong performance under strict single-pass, no-replay constraints.

---

### Official Review · Reviewer_F168 · 2025-11-01

**Soundness:** 2
**Presentation:** 1
**Contribution:** 1
**Rating:** 2
**Confidence:** 4

**Summary:**

The paper proposes Semantic Adapters (SAD), a parameter-efficient framework for online Domain-Incremental Learning (Domain-IL), where data arrives in a non-stationary, single-pass stream.

**Strengths:**

1-Parameter efficiency and fast speed.

2-Clear motivation.

**Weaknesses:**

[Major] 1-Incomplete paper submission: Captions of figures 1 and 2 are incomplete. The authors write long paragraphs for motivation, which has already been verified.

[Major] 2-The motivation is old: Finetuning all tasks with a unified head is not good, task-specific head is much better.

[Major] 3-Unsubstantiated Dynamic Routing: The proposed dynamic routing mechanism, which constitutes the paper’s main practical contribution, lacks sufficient justification and empirical support.

4-Questionable Value of SAD-LORA: The SAD-LORA variant dramatically increases the trainable parameters.

5-Missing Ablation on Language Alignment: The ablation of the Language Anchoring loss is missed.

**Questions:**

Please see the weakness.

---

> ### Author Response · Authors · 2025-11-26
>
> Thank you for your review. We appreciate your time and have incorporated all feedback to substantially revise the paper, which is now attached (with changes highlighted in blue). All changes and some new results are in the "global comment"
>
> - In the context of Domain-Incremental Learning (Domain-IL), the use of task-specific heads is actually a poor fit, which fundamentally justifies our semantic routing approach. Task-specific heads (i) require domain ID at inference, (ii) do not share statistics across domains, and (iii) cannot exploit cross-domain evidence for the same class. Further, as tasks/domains increase, we need to add more domain-wise classifiers and grow the network. In contrast, our semantic heads share parameters across all tasks by concept/meaning: reuse what “airplane” means across real/clip/sketch rather than relearning per domain, and also help this knowledge when a new unseen class "helicopter" class appears.
>
> - *Dynamic Routing* - Thank you for the feedback, we have added more details in the paper and also the appendix. Routing is required because online Domain-IL is ID-free: the model must decide, per input, which concept head to adapt without domain labels or replay. We combine three complementary signals that are each robust at different phases of training: (i) a language prior (group text prototypes) that is domain-invariant; (ii) running visual prototypes (feature means per concept); and (iii) head confidence (entropy from each head) that reflects how well a head already models the input. Further our contribution lies in the design concept of semantic or conceptual spaces to improve domain-adaptation in zero shot or online setting. More details on the language-based semantic clustering and training is also provided in the appendix.
>
> - We also add a new Plasticity-Stability analysis (Figure 7 in appendix) to show the trade-off.
> *Plasticity (New Learning)*: The semantic grouping forces the adapters to learn a generalized representation for a concept group . When a novel item is encountered in a new domain, the pre-trained adapter provides a robust initial setting, maximizing *forward transfer* and accelerating new learning.
> *Stability (Old Knowledge)*: By isolating the learning of different semantic concepts, the routing mechanism ensures that updates for 'animal' concepts do not interfere with 'tool' concepts, maintaining stability for past knowledge.
>
> - We provide two variants , SAD and SAD-LoRA, both of which are effective in its own ways and are still much better than fine-tuning. SAD trains only semantic heads and has the same #trainable parameters as task-wise classifiers. While SAD-LoRA is less efficient than SAD, it is still significantly more efficient than full fine-tuning. It activates targeted low-rank adapters only in the encoder tail We demonstrated in Table 5 that merely having more trainable parameters (e.g., Full Fine-Tuning) does not guarantee higher accuracy due to overwriting of information. SAD-LoRA, by contrast, uses the increased capacity (LoRA) selectively on the backbone, guided by the semantic routing heads.
> Example on VIT-B/16 -  trainables from ~400k (SAD) to ~1.2 (SAD-LoRA), which is <5% of a ViT-B/16 and ≪ full fine-tuning (e.g., 86.6M). Very detailed parameter count per network component is provided in Table 8.
>
> | Setting                          | Method        | Acc   | #P    |
> | -------------------------------- | ------------- | ----- | ----- |
> | Trainable Encoder                | PTM + 1 CLS   | 14.97 | 11.2M |
> |                                  | PTM + “N” CLS | 7.79  | 11.4M |
> | Frozen Encoder                   | PTM + 1 CLS   | 6.66  | 51.3k |
> |                                  | PTM + “N” CLS | 12.69 | 300k  |
> | Frozen Encoder + Semantic Groups | SAD           | 20.45 | 313k  |
> |                                  | SAD-LoRA      | 29.83 | 4.2M  |
>
>
> We also show results on adapter-based PTM-cl methods like DUCT (table in global comment) and SAD outperforms DUCT which retrains the whole network every task (accounting to 91M parameters)
>
> - Presentation fixes - We have improved the writing, captions and also changed few figures and added new analysis and ablations.
>
> We hope these clarifications address your concerns and increases your support for our work.

---

> > ### Comment · Reviewer_F168 · 2025-11-28
> >
> > Hi Author,
> >
> > Thanks for your rebuttal.
> >
> > I would like to increase my score to ***4*** based on your response.
> >
> > I still maintain the negative rate because of the limited contributions.
> >
> > Thank you.

---

### Author Response · Authors · 2025-11-26
**Global Comment**

We sincerely thank all the reviewers for their time and thoughtful feedback. We carefully considered every feedback and have substantially revised the paper—the story, writing, new results and all the ablations. The *updated paper* is attached and all the **changes can be seen in blue color**. The summary of all the changes are below,

- **Framing & clarity** - We have rewritten parts of the introduction/motivation for a crisper problem statement . We have also added more descriptive captions and expanded the explanations for all heatmaps and analytical figures.

- **Methodology** -  The main schematic (*Figure 3*) is revised completely to provide a clearer view of the framework. Dynamic routing is now explained in the main paper and also more in the Appendix. We also provide more explanation on the choice of "K" and how we perform semantic clustering using the language encoder.

- **Ablations**

- Language models - *Table7* - CLIP, BERT, and SBERT yield similar trends; showing that even lightweight language encoder suffices for relevant semantic clustering.
- Clustering - *Table 7* - k-means vs. GMM produces comparable results
- Loss function ablation - *Table 8* - Language anchoring consistently strengthens routing and cross-domain invariance; the separation term adds a modest, reliable margin between groups without inducing domain specialization.
- We report a detailed trainable-parameters in Table9, across architectures/datasets for full fine-tuning, for each component in SAD, and SAD-LoRA.

- **New results and analysis** - We include comparisons with more adapter-based and PTM-based Continual Learning methods in the online-setting premise in Table 4 and also shown below. DUCT and DCE have expensive adapter-based techniques that involve retraining of the whole network and multiple experts respectively. SAD outperforms with lower computational budget.
**Plasticity-Stability Analysis** - We include an explicit analysis of the model's plasticity (ability to learn new domains) and stability (ability to retain old knowledge), demonstrating how semantic routing effectively balances this critical trade-off (*Figure 7*).  SAD improves both metrics by organizing updates within concept-consistent subspaces, reducing interference and better preserving prior tasks.

| Method         | VIT-OfficeHome | # Train params |
| -------------- | ---------: | -------------: |
| DUCT [CVPR’24] |      75.90 |            91M |
| DCE [ICML’25]  |      64.00 |             3M |
| L2P [CVPR’22]  |      77.06 |           0.1M |
| SAD-Oracle     |      88.20 |              — |
| SADLoRA-Oracle |      89.24 |              — |
| SAD            |      80.14 |           443K |
| SADLoRA        |      81.80 |           1.9M |


**Summary** -

Real systems that continually face domain shift, cannot cache past data, and often see an instance only once. Yet much of Domain-IL still presumes multi-epoch training, replay buffers, or task/domain IDs. We start from the hypothesis that concept-level structure—not domain tags—should anchor learning: features that define “what it is” (wheels, wings, handles) are stable across styles, sensors, and weather. The contribution is a new intermediate level of knowledge organization: each head learns a generalized concept representation that is coarse enough to minimize parameter interference (stability) yet general enough to provide a strong prior for new classes maximizing positive forward transfer.  We provide two variants: one more efficient than the other, and the design is scalable (heads do not grow with tasks), efficient, and simple to deploy—properties that map well to real-world online settings.

We believe the feedback and the revisions have significantly improved the clarity, and completeness of our work. We hope our revisions and clarifications fully address your concerns and will motivate a positive reconsideration of the scores. We’re happy to discuss any remaining questions or provide further details as needed.

---

> ### Author Response · Authors · 2025-11-27
>
> - **Training and Inference Time** -  *Table 10 and 11*
> Both our variants train in a single epoch without replay and only update lightweight modules, so wall-clock training remains well under a few minutes.
>
> ResNet18 – DN4IL:
> | Method   | Train   | Inference |
> | -------- | ------- | --------- |
> | SAD      | 112.5 s | 0.95 ms   |
> | SAD-LORA | 201.8 s | 1.05 ms   |
>
> At inference, the time is in **"miliseconds"** making it suitable for real-time adaptations compared to other methods. For Example, DUCT method report "2000" seconds as their running time for VIT-B/16- Office home dataset. While both SAD and SADLora can perform training within 2 mins and inference per image under 7 miliseconds. Our results above demonstrate that semantic routing can be achieved with very low latency and training cost.
>
> ViT-B/16 – OfficeHome:
> | Method   | Train   | Inference |
> | -------- | ------- | --------- |
> | SAD      | 81.3 s  | 6.4 ms    |
> | SAD-LORA | 118.3 s | 7.1 ms    |

---

### Comment · Area_Chair_yYe7 · 2025-11-27
**Start discussion**

Hi all,

The authors have submitted their response to the initial reviews, and we now enter the discussion phase.

Please review the authors' response and the comments from other reviewers. Based on the rebuttal and discussion, please update your final score if appropriate.

We welcome and encourage further discussion as needed.

Thank you for your continued contributions.

Best regards,

Your AC.

---

### Author Response · Authors · 2025-12-01
**Final summary and note to the Area Chairs**

We sincerely appreciate everyone's efforts during this unusual review cycle and wish to highlight our methodology and the extensive improvements made during the rebuttal phase.

**Method & Impact:** We address the challenging problem of Online Domain-Incremental Learning (Online DIL), where models must adapt to severe distribution shifts in a single pass without memory and replay. Our core innovation, **Semantic Adapters (SAD)**, replaces traditional domain-specific adaptation with a **novel memory organization**. We leverage a frozen language model to cluster samples into stable Semantic Concepts. A Semantic Routing mechanism dynamically assigns inputs to lightweight, concept-specific adapters, ensuring that learning updates are isolated to relevant concepts (stability) while allowing related classes to share capacity (forward transfer). We also propose dynamic semantic routing (and not use any extra information such as task or domain-ID during inference) to assign samples to the semantic adapters by using three complementary signals that are each robust at different phases of training. We provide two variants—SAD and SAD-LoRA. SAD-LoRA is less efficient than SAD and was shown to demonstrate that the semantic routing structure remains robust and effective even when we introduce small, targeted low-rank updates in the encoder tail to handle large domain shifts. We achieve State-of-the-art performance on 2 architectures and 3 datasets, against 10 different SOTA methods, the most recent being a paper from ICML'25 .

**Rebuttal Summary** : We have revised the manuscript extensively (all changes in blue) with all the feedback. Two reviewers explicitly increased their scores after reading the rebuttal; the other two did not get time to respond after our final detailed reply on Nov 27.
The main concerns from reviewers were on ablations, clarity, and novelty. We have added new results against recent adapter-based baselines (table also in *"Global comment"* below and a plasticity–stability analysis, as well as measured training and inference times. We have provided ablations on semantic groups "K", language encoders, clustering, loss functions and hyper-parameters. We also rewrote the introduction, added new figures, schematics and explanations and revised the manuscript significantly.

The hesitation about **novelty** because our method uses pre-trained models (PTM), and low rank matrices, we respectfully feel, is an incomplete view of novelty in the current landscape. PTM based CL is a new area - as modern CL focuses on efficiently adapting models and explore different prompts/low rank matrices for fine-tuning; what distinguishes methods is how they learn, add experts/prompts and update knowledge under realistic constraints. Most works also focus on Class-IL and non-online settings. Our work is a new, system-level formulation that organizes memory by semantic concepts and fundamentally redefines knowledge isolation from domain-specific to concept-specific, a **paradigm shift for Domain-IL**. It further offers an extremely practical solution for modern continual learning—it is simple, scalable, and extremely efficient (milliseconds latency, minimal parameters) - Ex. in VIT we use only *443K trainable parameters compared to 86M in VIT and infer in ~6 miliseconds*.

Beyond empirical gains, the contribution is a practically useful recipe: starting from any frozen PTM, one can leverage a small LLM to build semantic groups and obtain robust, real-time online domain adaptation. We believe this combination of conceptual clarity, strong empirical support, and deployment-ready efficiency makes the work a valuable addition to both the academic and industrial communities

---

### Note · Program_Chairs · 2026-01-17
**Submission Desk Rejected by Program Chairs**

The following references in this submission do not refer to real documents and/or have major errors in bibliographic information:

 Pietro Buzzega, Matteo Boschini, Andrea Porrello, and Simone Calderara. Der++: Improved deep episodic replay for continual learning. In CVPR, 2020.